# Knowledge Retention in Continual Model-Based Reinforcement Learning

**Haotian Fu** [* 1]   **Yixiang Sun** [* 1]   **Michael Littman** [1]   **George Konidaris** [1]

## Abstract

We propose DRAGO, a novel approach for continual model-based reinforcement learning aimed at improving the incremental development of world models across a sequence of tasks that differ in their reward functions but not the state space or dynamics. DRAGO comprises two key components: *Synthetic Experience Rehearsal*, which leverages generative models to create synthetic experiences from past tasks, allowing the agent to reinforce previously learned dynamics without storing data, and *Regaining Memories Through Exploration*, which introduces an intrinsic reward mechanism to guide the agent toward revisiting relevant states from prior tasks. Together, these components enable the agent to maintain a comprehensive and continually developing world model, facilitating more effective learning and adaptation across diverse environments. Empirical evaluations demonstrate that DRAGO is able to preserve knowledge across tasks, achieving superior performance in various continual learning scenarios.

## 1. Introduction

Model-based Reinforcement Learning (MBRL) aims to enhance decision-making by developing a world model that captures the underlying dynamics of the environment. A robust world model allows an agent to predict future states, plan actions, and adapt to new situations with minimal real-world trial and error. For MBRL to be effective in dynamic, real-world applications, the world model must incrementally learn and adapt, continually integrating new information as the agent encounters diverse environments and tasks.

Imagine an agent initially exploring a small, confined part of a complex world, like a home-assistance robot learning to navigate a kitchen. At first, the robot masters dynamics specific to that environment, such as avoiding countertops and maneuvering around chairs. When deployed to a new household, it must adapt to unfamiliar layouts while retaining its understanding of prior environments. Over time, as the robot encounters diverse settings—homes with varying furniture arrangements, hospitals with strict privacy constraints, or factories with evolving machinery—it must incrementally integrate new dynamics without forgetting earlier knowledge. This process aligns with the principles of *continual learning*, where agents progressively acquire new skills across tasks while preserving past experiences (Lange et al., 2022). However, real-world constraints often prohibit storing raw interaction data from prior tasks due to: **1. Storage limitations**: Robots or embodied agents cannot indefinitely retain growing datasets (Hadsell et al., 2020). **2. Privacy regulations**: Healthcare or service robots handling sensitive data may be restricted from archiving task-specific interactions (Kemker et al., 2018). **3. On-device deployment**: Deployed AI systems (e.g., smartphones) often rely on pre-trained models where the original training data is proprietary or privacy-sensitive.

In principle, continual MBRL would enable agents to learn a generalizable world model supporting a universal set of tasks. If data from all previous tasks are available, multitask learning strategies (Fu et al., 2022) could solve this by leveraging shared dynamics. However, in storage- or privacy-constrained settings, agents **lack access to prior task data**, rendering such approaches infeasible. In practice, we actually always have bounded storage. If we consider the hardest case, where no previous data is available: as shown in Figure 1 and our experiments, naive MBRL suffers catastrophic forgetting, overwriting critical dynamics (e.g., a robot forgetting kitchen layouts after learning living rooms). To address this, we need strategies that retain essential knowledge without storing past data, enabling agents to build increasingly complete world models across tasks.

Specifically, we propose DRAGO, a novel continual MBRL approach designed to address catastrophic forgetting and incomplete world models in the absence of prior task data. DRAGO consists of two key components: Synthetic Experience Rehearsal and Regaining Memories Through Exploration. *Synthetic Experience Rehearsal* uses a continually learned generative model to enable the agent to simulate and learn from synthetic experiences that resemble those

---

*Equal contribution   [1]Brown University.   Correspondence to: Haotian Fu <hfu@cs.brown.edu>, Yixiang Sun <ysun133@cs.brown.edu>.

*Proceedings of the 42nd International Conference on Machine Learning*, Vancouver, Canada. PMLR 267, 2025. Copyright 2025 by the author(s).

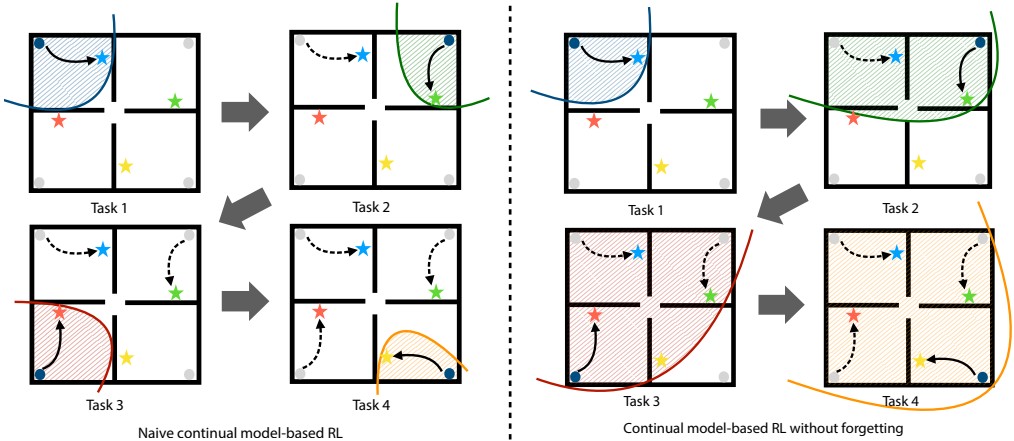

*Figure 1.* Comparison between the world model learned by naive continual MBRL and MBRL without forgetting. Each task requires the agent to move from the corner of one room to a specific point in the same room. Shaded areas represent the world model's coverage after finishing each task. Naively continually training MBRL (*Left*) tends to suffer the catastrophic forgetting problem—the agent forgets almost everything about the first room after training in the second room (our experimental results support this claim). Our project identifies a method (*Right*) that helps the world model preserve the knowledge of previous tasks even when the old data is no longer available.

from prior tasks. This process allows the agent to synthesize representative transitions that resemble prior experience, reinforcing its understanding of previously learned dynamics without requiring access to past data. In the *Regaining Memories Through Exploration* component, we introduce an intrinsic reward mechanism that encourages the agent to actively explore states where the previous transition model performs well. This exploration bridges the gap between tasks by discovering connections within the environment, leading to a more comprehensive and cohesive world model. To sum up, we make the following contributions:

1. We introduce DRAGO, a new approach for continual model-based reinforcement learning that addresses catastrophic forgetting while incrementally learning a world model across sequential tasks without retaining any past data.

2. We propose a generative replay mechanism that synthesizes "old" transitions using a learned generative model alongside a frozen copy of the previously trained world model.

3. We design an intrinsic reward signal that nudges the agent toward revisiting states that the old model explained well—effectively "reconnecting" current experiences with previously learned transitions.

4. Through experiments on MiniGrid and DeepMind Control Suite domains, we show that DRAGO substantially improves knowledge retention compared to standard continual MBRL baselines and achieves higher transfer performance, allowing faster adaptation to entirely new (but related) tasks. Code is available at https://github.com/YixiangSun/drago.

## 2. Background

In reinforcement learning, an agent interacts with an environment modeled as a Markov Decision Process (MDP). An MDP is defined by a tuple $< \mathcal{S}, \mathcal{A}, T, r, \gamma >$, where $\mathcal{S}$ is the state space, $\mathcal{A}$ is the action space, $T(s' \mid s, a)$ represents the transition dynamics, $r(s, a)$ is the reward function, and $\gamma \in [0, 1)$ denotes the discount factor. Throughout this paper, $T$ is the parametric transition model (the learned dynamics predictor) and $p$ denotes a probability or distribution.

In continual model-based reinforcement learning, the agent is presented with a sequence of tasks $\mathcal{T}_1, \mathcal{T}_2, \ldots, \mathcal{T}_n$. We assume the agent knows when the task switches. Each task $\mathcal{T}_i$ is associated with its own MDP, $\mathcal{M}_i = (\mathcal{S}, \mathcal{A}, T, r_i, \rho_i, \gamma)$, where $r_i(s, a)$ is the task-specific reward function, and $\rho_i(s)$ denotes the initial state distribution for task $\mathcal{T}_i$. Importantly, all tasks share the same transition function $T(s' \mid s, a)$, which defines the probability of reaching state $s' \in \mathcal{S}$ from state $s \in \mathcal{S}$ after taking action $a \in \mathcal{A}$. In this paper, we consider the case where, in each task, the agent tends to be exposed to distinct aspects of the transition dynamics and different termination states.

The objective in continual MBRL is to efficiently solve the sequence of tasks, while learning a world model $T_\psi(s' \mid s, a)$, parameterized by $\psi$, that captures the shared dynamics across all tasks, allowing the agent to adapt to the task-specific objectives defined by $r_i$ and $\rho_i$. A challenge arises because, during training on a new task $\mathcal{T}_i$, the agent in our setting only has access to the replay buffer $\mathcal{B}_i = \{(s, a, s')\}$.

# 3. DRAGO

The central question in this paper is: how do we aggregate the knowledge from previous tasks and learn a increasingly complete world model without forgetting, while trying to solve a sequence of tasks using MBRL? As shown in previous works (Fu et al., 2022), the agent can easily learn a general world model in a multitask/meta-learning way as long as the access to previous tasks' memories is given. Thus, a straightforward way is to figure out an approach that is able to **regain** the old memories that had to be discarded. We propose DRAGO, a continual MBRL approach is composed of two main components: dreaming and rehearsing old memories while training on new tasks (§3.1), and regaining memories via actively exploration (§3.2). Then we introduce the overall algorithm in §3.3.

## 3.1. Synthetic Experience Rehearsal

To help the agent retain knowledge from previous tasks without direct access to past data, we introduce a method called *Synthetic Experience Rehearsal*. This approach enables the agent to internally generate and learn from synthetic experiences that resemble those from prior tasks, effectively reinforcing its understanding of the environment's dynamics and mitigating catastrophic forgetting.

The concept of *Synthetic Experience Rehearsal* draws inspiration from how humans and animals replay and consolidate memories during sleep (Wilson & McNaughton, 1994). Just as dreaming allows for the consolidation of memories and learning in biological systems, our method helps the agent retain and reinforce knowledge of previous dynamics by generating and learning from synthetic experiences. Imagine a robot that has navigated through several rooms in a building. As it progresses to new rooms, it may begin to forget the layouts and navigation strategies of earlier ones due to limited memory capacity and the inability to revisit those rooms. By internally generating and rehearsing synthetic experiences that mimic its interactions in earlier rooms, the robot can maintain and reinforce its knowledge of how to navigate them. This internal rehearsal helps it integrate past experiences with new ones, ensuring a more comprehensive understanding of the entire environment.

Our method leverages a generative model (which is also continually learned) to produce synthetic data that aids in training the dynamics model, thereby preventing forgetting of previously learned dynamics. Note that for real-world tasks, **retaining the model (neural nets) usually costs much less than retaining all the training transitions**, especially when the task number grows larger and larger.

Specifically, we employ a generative model $G$ that encodes and decodes both states and actions, capturing the joint distribution of state-action pairs encountered in previous tasks.

Including actions is crucial, especially in continuous action spaces where randomly sampled actions may not correspond to meaningful behaviors. Throughout the continual learning process, we also keep one copy of the **old** world model learned after finishing the last task (**one for all previous tasks, not one for each**). Then after generating the state-action pair, we feed it into this frozen old model $T_{\text{old}}$ and generate a synthetic next state. The synthetic data used for training the model is generated through the following steps:

$$\hat{s}' = T_{\text{old}}(\hat{s}, \hat{a}), \ (\hat{s}, \hat{a}) \sim p_G(s, a; \theta), \tag{1}$$

where $p_G(s, a; \theta)$ is the distribution modeled by the generative model $G_\theta$ with parameters $\theta$. $T_{\text{old}}$ is the frozen old world model, capturing the dynamics up to a previous task.

We can express the likelihood of the entire dataset, including both real data $\mathcal{D}_i$ for current task $\mathcal{T}_i$ and synthetic data $\hat{\mathcal{D}}$, given the parameters $\psi$ and $\theta$, as follows:

$$
p(\mathcal{D}_i, \hat{\mathcal{D}} \mid \psi, \theta) = \\
\prod_{(s,a,s') \in \mathcal{D}_i} p(s' \mid s, a; \psi) \prod_{(\hat{s}, \hat{a}, \hat{s}') \in \hat{\mathcal{D}}} p_G(\hat{s}, \hat{a}; \theta) p(\hat{s}' \mid \hat{s}, \hat{a}; \psi),
\tag{2}
$$

where $p(s' \mid s, a; \psi)$ is the likelihood of observing $s'$ given $s$ and $a$ under the transition model $T_\psi$, $p_G(\hat{s}, \hat{a}; \theta)$ is the likelihood of generating synthetic state-action pairs from the generative model $G_\theta$. This joint likelihood captures the dependencies of the synthetic data on both the generative model parameters $\theta$ and the frozen transition model $T_{\text{old}}$.

The posterior distribution over the transition model parameters $\psi$ and the generative model parameters $\theta$ is given by Bayes' theorem:

$$p(\psi, \theta \mid \mathcal{D}_i, \hat{\mathcal{D}}) \propto p(\mathcal{D}_i, \hat{\mathcal{D}} \mid \psi, \theta) \, p(\psi) \, p(\theta), \tag{3}$$

where $p(\psi)$ and $p(\theta)$ are the prior distributions over the parameters.

Taking the negative logarithm of the posterior (and ignoring constants independent of $\psi$ and $\theta$), we obtain the joint loss function:

$$
\mathcal{L}_{\text{total}}(\psi, \theta) = -\log p(\mathcal{D}_i, \hat{\mathcal{D}} \mid \psi, \theta) - \log p(\psi) - \log p(\theta) \\
= -\log p(\psi) - \log p(\theta) - \underbrace{\sum_{(s,a,s') \in \mathcal{D}_i} \log p(s' \mid s, a; \psi)}_{\text{Loss on current task data}} \\
- \underbrace{\sum_{(\hat{s}, \hat{a})} \log p_G(\hat{s}, \hat{a}; \theta) - \sum_{(\hat{s}, \hat{a})} \log p(\hat{s}' \mid \hat{s}, \hat{a}; \psi)}_{\text{Synthetic data likelihood}}.
\tag{4}
$$

The dynamics model is trained by minimizing the prediction loss over the combined dataset:

$$
\mathcal{L}_{\text{dyn}}(\psi) = \mathbb{E}_{(s,a,s') \sim \mathcal{D}_i} \left[ \|s' - T_i(s, a; \psi)\|^2 \right] \\
+ \lambda \mathbb{E}_{(\hat{s}, \hat{a}) \sim p_G(s, a; \theta)} \left[ \|T_{\text{old}}(\hat{s}, \hat{a}) - T_i(\hat{s}, \hat{a}; \psi)\|^2 \right],
\tag{5}
$$

where $\lambda$ is a weighting factor controlling the importance of the synthetic data loss. While this enables the agent to learn from synthetic old experience, the generative model itself (minimizing $-\sum \log p_G(\hat{s}, \hat{a}; \theta)$ in Eqn 4) also requires accumulating the knowledge of different tasks as the training goes on. Retaining such a generative model for every task will also introduces huge additional cost.

**Continual learning for the generative model.** To prevent forgetting within the generative model itself, we adopt a continual training strategy. We generate synthetic state-action pairs using the previous generative model $G_{i-1}$:

$$(\tilde{s}, \tilde{a}) = G_{i-1}(\tilde{z}), \tilde{z} \sim p(z),$$

and combine these with real data from the current task to form the training dataset for the new generative model: $\mathcal{D}_{\text{gen}} = \mathcal{D}_i \cup \tilde{\mathcal{D}}$, where $\tilde{\mathcal{D}} = \{(\tilde{s}, \tilde{a})\}$. The new generative model $G_i$ — we use Variational AutoEncoder (VAE) (Kingma & Welling, 2014) — is then trained by minimizing the loss over $\mathcal{D}_{\text{gen}}$:

$$\begin{aligned} \mathcal{L}_{\text{gen}}(\theta_i) =& \mathbb{E}_{(s,a) \sim \mathcal{D}_{\text{gen}}} \Big[ -\mathbb{E}_{z \sim q_{\theta_i}(z|s,a)} \left[ \log p_{\theta_i}(s, a \mid z) \right] \\ &+ \text{KL}\left( q_{\theta_i}(z \mid s, a) \,\|\, p(z) \right) \Big]. \end{aligned} \tag{6}$$

This continual learning procedure ensures that the generative model retains its ability to produce state-action pairs representative of all previous tasks.

Our method is general and can be applied with other types of generative models. Additionally, integrating more sophisticated generative models, such as diffusion models, could further enhance the quality of synthetic experiences and improve knowledge retention in high-dimensional environments. We leave this for future work.

### 3.2. Regaining Memories Through Exploration

While generating synthetic data via a generative model helps mitigate forgetting, it may not fully capture the richness of real experiences and it is subject to model error. In the meantime, to eventually build a complete world model, we would like to find a way that can **"connect" knowledge gained from different tasks if they are disjoint**. Thus, to further enhance the agent's retention of prior knowledge and make the world model more complete, we propose an intrinsic reward mechanism that encourages the agent to actively explore states where the previous transition model performs well, effectively "regaining" forgotten memories through real interaction with the environment, and fill in the gap between knowledge of different tasks.

Our approach is inspired by the need to complement the generation-based rehearsal method with actual exploration that bridges the **gap** between different tasks. The generative model can produce states from prior tasks, but these imagined states might not be naturally encountered or connected

within the current task. Consider the earlier example of a robot exploring different rooms within a building. The method introduced in the last section can generate imagined states from previously visited rooms, but without actual exploration, the robot might not find the doorways or corridors connecting these rooms to its current location. Our intrinsic reward incentivizes the robot to search for these connections, enabling it to discover pathways that link the new room to the old ones. Without exploring the actual environment to find these connections, the agent's world model remains fragmented, lacking a cohesive understanding of how different regions relate. To overcome this, we propose an intrinsic reward that guides the agent to:

- **Revisit Familiar States**: Encourage exploration of states where the previous model $T_{i-1}$ predicts accurately, indicating familiarity from earlier tasks.

- **Discover New Connections**: Incentivize the agent to find paths that connect current and previous task environments, enriching the world model's completeness.

- **Balance Learning Dynamics**: Deter the agent from spending excessive time in regions where the current model $T_i$ already performs well.

Specifically, during training on task $\mathcal{T}_i$, we introduce an intrinsic reward $r_{\text{cont}}^i$ designed to guide the agent towards states that are familiar to the previous transition model $T_{i-1}$ (trained and froze after task $\mathcal{T}_{i-1}$) but less familiar to the current model $T_i$. The intrinsic reward is defined as:

$$\begin{aligned} r_{\text{cont}}^i(s_t, a_t, s_{t+1}) :=& \sigma\left(-\log|T_{i-1}(s_t, a_t) - s_{t+1}|\right) \\ &- \alpha \cdot \sigma\left(-\log|T_i(s_t, a_t) - s_{t+1}|\right), \end{aligned} \tag{7}$$

where $\sigma$ denotes the sigmoid function, and $\alpha$ is a weighting coefficient that balances the two terms.

Intuitively the first term assigns higher rewards when the previous transition model $T_{i-1}$ predicts the next state $s_{t+1}$ accurately. This incentivizes the agent to revisit states that were well-understood in previous tasks. The second term penalizes the agent for visiting states where the current model $T_i$ already has low prediction error. This encourages the agent to explore less familiar areas to improve the current model's understanding.

By actively exploring and connecting different regions, the agent's world model becomes more comprehensive, capturing the dynamics across tasks more effectively. Revisiting familiar states reinforces prior knowledge, reducing the tendency of the model to forget previously learned information. This approach complements the synthetic data generation in Section 3.1 by providing actual experience that reinforces the agent's knowledge.

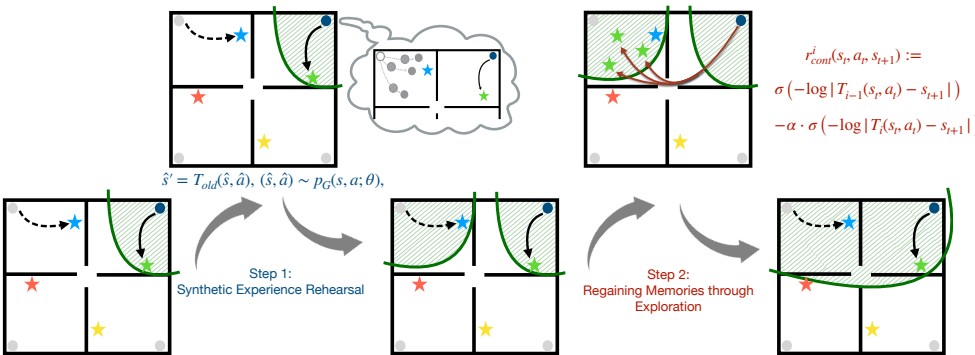

$\hat{s}' = T_{old}(\hat{s}, \hat{a}),\ (\hat{s}, \hat{a}) \sim p_G(s, a; \theta),$

Step 1:
Synthetic Experience Rehearsal

Step 2:
Regaining Memories through
Exploration

$r^i_{cont}(s_t, a_t, s_{t+1}) :=$

$\sigma\left(-\log|T_{i-1}(s_t, a_t) - s_{t+1}|\right)$

$-\alpha \cdot \sigma\left(-\log|T_i(s_t, a_t) - s_{t+1}|\right)$

*Figure 2.* The two-step process of how DRAGO retain and aggregate the knowledge learned from prior tasks for the world model. Step 1 involves *Synthetic Experience Rehearsal*, where synthetic state-action pairs are generated from the previous tasks' generative model $G_{i-1}(z)$, and next states $\hat{s}'$ are predicted using the previous transition model $T_{i-1}$. Step 2 introduces *Regaining Memories through Exploration*, where an intrinsic reward $r^i_{cont}$ encourages the agent to explore states where the previous transition model $T_{i-1}$ performs well, while penalizing states that the current model $T_i$ already predicts accurately. Together, these components allow the agent to retain and transfer knowledge across tasks.

### 3.3. Overall Algorithm

We implement DRAGO on top of TDMPC (Hansen et al., 2022) and the overall algorithm is described in Algorithm 1. Compared to regular TDMPC algorithm, we additionally train an encoder and decoder for the state-action pair as part of the generative model in §3.1. To integrate the intrinsic reward for regaining memories proposed in §3.2, we train an additional reward model, value model, and policy as a "reviewer" that aims to maximize the cumulative intrinsic reward, besides the original "learner" that aims to maximize the cumulative environmental reward. Note that the reviewer and the learner share the same world model, which is also trained using data from both.

During the inference step, DRAGO leverages Model Predictive Path Integral (Williams et al., 2015) as the planning method. Given an initial state and task $\mathcal{T}_i$, DRAGO samples $N$ trajectories with the world model $T_i$ and estimates the total return $J_\tau$ of each sampled trajectory $\tau$ as:

$$J_\tau := \mathbb{E}_\tau \left[ \sum_{t=0}^{H-1} \gamma^t R_{s_t, a_t} + \gamma^H Q(s_H, a_H) \right],\ s_{t+1} \sim T_i(s_t, a_t; \psi), \tag{8}$$

where $Q(\cdot)$ is the learned value function. Then a trajectory with the highest return is picked and the agent will execute the first action in the trajectory.

During training, the dynamics model and the generative model are trained together with the reward&value prediction of the learner and reviewer. At the beginning of each new task, **for each new test task, we randomly initialize the reward, policy and value models and reuse only the world model (dynamics).**. Moreover, unlike TDMPC, the gradients from updating Q function and reward model are detached for updating the dynamics model in DRAGO. More implementation details can be found in the appendix.

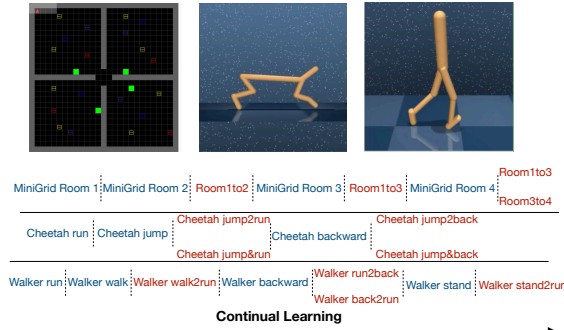

*Figure 3.* Visualization of the evaluated domains. Task names in Blue denote the continual **training** tasks; Task names in Red denote the **test** tasks. We train and test all the tasks in the order of left to right as in the figure. E.g., we train the cheetah agent in the order of run, jump and backward. And after training on jump, we test on jump2run and jump&run.

## 4. Experiments

We evaluated DRAGO on three continual learning domains. For each domain, we let the agent train on a sequence of tasks, where the tasks share the same transition dynamics but different reward functions. Although the transition dynamics are the same, the training tasks are designed in a way such that to solve each task only part of the state space's transition dynamics needs to be learned and different tasks involve learning transition dynamics corresponding to different parts of the state space **with a small overlap**. We evaluate the agent's continual learning performance on test tasks by measuring the agent's training performance on them, using the retained world model as an initiation. The test tasks requires the combination of knowledge from more than one previously learned tasks. For example, to better transfer on *Cheetah jump2run* the agent is expected to still remember the knowledge learned in *Cheetah run* even after continual training on *Cheetah jump*. These transfer tasks

are designed to test the agent's ability to retain knowledge from previous tasks, as solving them requires understanding multiple tasks.

**MiniGrid.** We evaluated the performance of DRAGO in the MiniGrid (Chevalier-Boisvert et al., 2023) domain using a sequence of four tasks, each set in one of the four rooms of a $27 \times 27$ gridworld. In each task, the agent starts from a fixed corner of one room, with the objective of reaching a specified goal position within that room. The obstacles vary across tasks and the agent can only access other rooms by passing through a door located at the center of the gridworld, which creates a bottleneck that the agent must learn to navigate effectively in transfer tasks. Each task requires exploring a small and mostly non-overlapping portion of the world, ensuring that knowledge from one task does not directly overlap with others. To assess transfer performance, we evaluated the models learned at different stages of the continual learning process (i.e., after completing 2, 3, and 4 tasks). The evaluation was conducted on four new tasks that require the agent to move between different rooms (e.g., start in room 1 and move to the goal position in room 2). The tasks are designed such that solving them requires understanding multiple rooms.

**Deepmind Control Suite.** We also evaluated the performance of DRAGO in the Cheetah and Walker domains from the Deepmind Control Suite (Tassa et al., 2018). For each domain, we define a sequence of tasks that share the same dynamics but with different task goals, which requires the agent to learn different parts of the state space of dynamics. Similarly, to assess transfer performance, we evaluated the models learned at different stages of the continual learning process. The evaluation was conducted on several new tasks that require the agent to quickly change to different locomotion modes from another mode (jump, run, etc.), except for two tasks: *jump and runforward* & *jump and runbackward*, where the agent will get the maximum reward if it runs forward/backward and jumps at the same time.

We compared to baselines including: Training **TDMPC from scratch** for each task, **continual TDMPC**, where we initialize the world model with the one learned in the previous task at the beginning of the new task and train it with the task reward, and **EWC**, a regularization-based continual learning method as we introduced in the related work section. We use TDMPC as the base model-based reinforcement learning (MBRL) algorithm for all the baselines. More experimental results can be found in appendix E & D.

### 4.1. Qualitative Results

In Figure 4, we also visualize the prediction accuracy of the learned world models across the whole gridworld, comparing just naively continually training TDMPC and our method. The prediction score is calculated based on the states predictions' mean square error (MSE). The results are aligned with our intuition. Without other counter-forgetting techniques, world models easily forget almost everything learned in previous tasks and are only accurate in the transition space related to the current task. By contrast, DRAGO is able to retain most of the knowledge learned in previous tasks and have a increasingly complete world model as training continues, leading to the performance gain on new tasks shown in Figure 5. Note that DRAGO's performance without *Synthetic Experience Rehearsal* (so only has the *Regaining Memories Through Exploration* Component) drops a bit compared to the full version, but it still exhibits better knowledge retention to some extent in post-task3 and post-task4, compared to naive continual TDMPC. As we also show in the ablation study, combining two components of DRAGO eventually achieves the best overall performance.

### 4.2. Overall Performance

As shown in Figure 5, we find that the proposed method DRAGO achieves the best overall performance compared to all the other approaches across three domains. The results demonstrate its advantage in continual learning settings by effectively retaining knowledge from previous tasks and transferring it to new ones. We can also see that naively continual Model-based RL may suffer from severe plasticity loss: Continual TDMPC constantly performs worse than learning from scratch baseline. Equipped with EWC, it can achieve better overall performance but still not as good as DRAGO. But DRAGO does not fully alleviate the plasticity loss, in *Cheetah Jump and runbackward* (Last plot in the mid row of Figure 5), learning from scratch still has the best performance, but we can see that DRAGO still improves a lot compared to Continual TDMPC.

### 4.3. Ablation Study

This section evaluates the essentiality of DRAGO's components. Specifically, we evaluate DRAGO's performance without *Synthetic Experience Rehearsal* and *Regaining Memories Through Exploration* (reviewer) separately in four transfer tasks of Cheetah and MiniGrid. As we show in Figure 7, while *DRAGO w/o. Rehearsal* achieves similar performance with the full version in *Cheetah-jumpandrunforward*, the full DRAGO still has the best overall performance across domains. If we compare the performance with Continual TDMPC shown in Figure 5, one single component of DRAGO consistently improves continual learning performance. These results highlight the complementary roles of both components and demonstrate that each contributes significantly to mitigating forgetting and enhancing transfer capabilities in continual model-based RL settings.

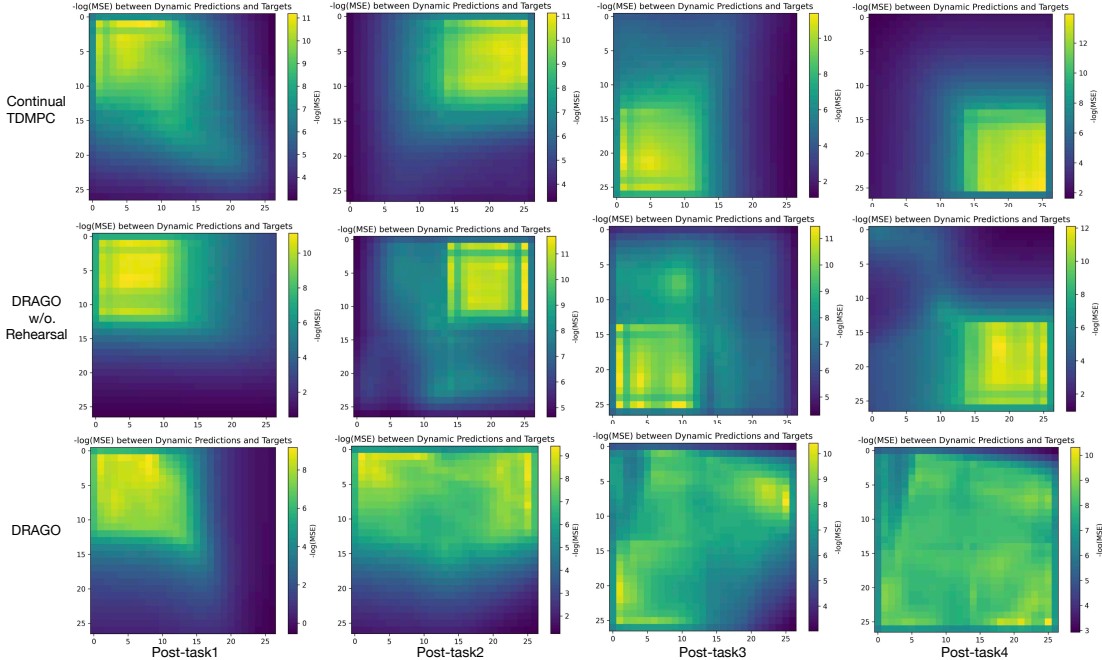

*Figure 4.* **DRAGO reduces the catastrophic forgetting from previous tasks**, illustrated bt the prediction score of the learned world models across the entire gridworld after each task. Light color indicates higher prediction accuracy. The heatmaps compare the performance of naive continual training of TDMPC (top row), DRAGO without *Synthetic Experience Rehearsal* (mid row), with our proposed full DRAGO method (bottom row) after Tasks 1 to 4. The results show that continual MBRL suffers from significant forgetting, maintaining accuracy only in regions relevant to the current task, whereas DRAGO effectively retains knowledge from previous tasks.

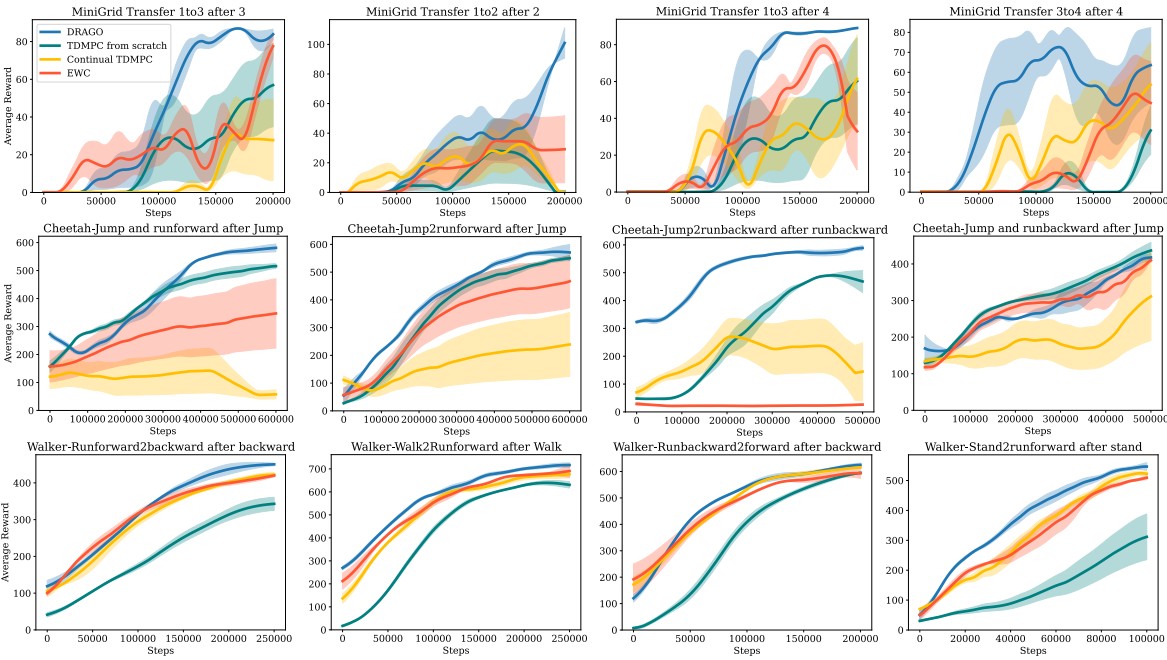

*Figure 5.* We evaluate the continual learning transfer performance on 12 tasks (3 domains, 4 tasks each) that are not seen during the agent's previous training. Each plot corresponds to a single test task, and the agent's performance is tracked as it learns that task from scratch, using the retained world model. For each test task of MiniGrid, the agent starts in one room and have to move to the goal in another room. E.g., *Transfer 3to4 after 4* means that after sequentially training on four tasks, the agent is tested on a new task where it starts in room 3 and the target position is in room 4. For each test task of Cheetah & Walker, the agent has to start from a state in one locomotion mode and the goal is to switch to another mode. E.g., *Jump2runforward after Jump* means that after training on Cheetah-Jump, the agent is tested on a new task where it starts in one state of the jumping mode, and the goal is to run forward.

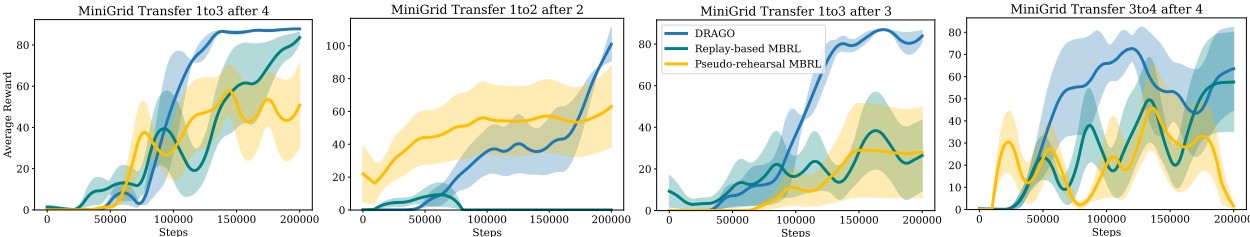

*Figure 6.* Comparison of DRAGO with (1) Replay-based MBRL, which stores a bounded replay buffer for past tasks (using the same memory size as DRAGO's generative model), and (2) Pseudo-rehearsal MBRL (Ketz et al., 2019), which generates old data via a pretrained VAE model, without continual training of the generative model as well as our intrinsic reward mechanism. Each plot shows Minigrid transfer performance (average return vs. environment steps) on a new task after sequentially training on earlier tasks. DRAGO consistently achieves higher or faster-improving returns, suggesting stronger knowledge retention and quicker adaptation to the transfer tasks.

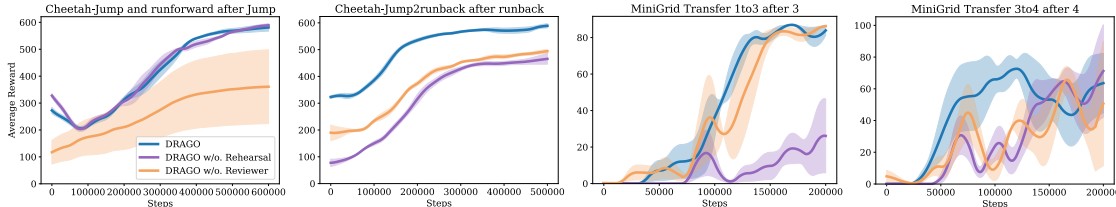

*Figure 7.* Ablation study results on four transfer tasks in the Cheetah and MiniGrid domains, comparing the performance of DRAGO without individual components (*Synthetic Experience Rehearsal* and *Regaining Memories Through Exploration*) to the full method. While removing *Rehearsal* results in competitive performance in the *Cheetah-jumpandrunforward* task, the full version of DRAGO achieves superior overall performance across all tasks.

### 4.4. Few-shot Transfer Performance

We also evaluated the agent's few-shot transfer performance during the continual learning process and compared the results of DRAGO with the other baselines. The setting is useful and common in real world tasks, especially for robotics, where the number of steps to interact with the environment is limited. Specifically, for each test task in Cheetah and Walker domains, we let the agent train by interacting with the environment for only 20 episodes and evaluate its average cumulative reward after training. As shown in Table 1, DRAGO outperforms the other baselines in 6 out of 8 tasks. In the two tasks where DRAGO does not outperform, it remains competitive, highlighting its robustness and efficiency in continual learning scenarios.

## 5. Related Work

Continual reinforcement learning (CRL) aims to develop agents that can learn from a sequence of tasks, retaining knowledge from previous tasks while efficiently adapting to new ones (Khetarpal et al., 2022; Abel et al., 2023; Anand & Precup, 2023; Baker et al., 2023). Many recent papers investigate the plasticity loss in continual learning (Lyle et al., 2023; Abbas et al., 2023; Dohare et al., 2024). This paper focuses more on how we better retain and aggregate knowledge learned from previous tasks in Continual MBRL, which is related to another central challenge in CRL,

catastrophic forgetting, where learning new tasks causes the agent's performance on earlier tasks to degrade due to the overwriting of important knowledge (McCloskey & Cohen, 1989). To address catastrophic forgetting, several strategies have been proposed: Regularization-Based Methods (Kirkpatrick et al., 2016; Li & Hoiem, 2016; Zenke et al., 2017; Nguyen et al., 2017; Yu et al., 2020a): these approaches introduce constraints during training to prevent significant changes to parameters important for previous tasks. Elastic Weight Consolidation (EWC) (Kirkpatrick et al., 2016) is a prominent example that uses the Fisher Information Matrix to estimate parameter importance and penalize updates accordingly. However regularization-based methods often struggles in practice, especially in reinforcement learning scenarios, due to challenges in accurately estimating parameter importance and scalability issues with large neural networks (Huszár, 2017; Farquhar & Gal, 2018). Replay-Based methods (Riemer et al., 2019; Rolnick et al., 2019; Oh et al., 2022; Henning et al., 2021; Lampinen et al., 2021): these methods typically assume unbounded storage, which is usually not feasible in practice. Our work is therefore focused on the hardest case — alleviating the catastrophic forgetting problem and learn a complete world model without prior data. Generative replay-based methods (Shin et al., 2017; Triki et al., 2017; Rao et al., 2019) share some similar high-level idea with our work. However, none of them has been applied on MBRL or World Models. In terms of Continual MBRL specifically, Fu et al. (2022) show that the

| Average Reward | DRAGO | EWC | Continual TDMPC | Scratch |
|---|---|---|---|---|
| *Cheetah jump2run* | **106.78 ± 32.01** | 54.72 ± 62.72 | 93.96 ± 39.29 | 26.54 ± 2.67 |
| *Cheetah jump&run* | **248.92 ± 15.38** | 156.98 ± 99.68 | 128.58 ± 100.14 | 182.77 ± 28.58 |
| *Cheetah jump2back* | **331.85 ± 11.05** | 29.93 ± 7.15 | 73.98 ± 38.45 | 45.15 ± 4.92 |
| *Cheetah jump&back* | **147.30 ± 34.29** | 117.92 ± 1.20 | 140.82 ± 28.00 | 129.75 ± 20.44 |
| *Walker walk2run* | **332.38 ± 20.07** | 287.02 ± 37.80 | 229.14 ± 33.71 | 52.11 ± 3.41 |
| *Walker run2back* | 145.98 ± 17.96 | **150.19 ± 2.77** | 128.56 ± 9.47 | 60.49 ± 9.40 |
| *Walker back2run* | 229.79 ± 9.77 | **254.09 ± 70.29** | 241.39 ± 42.64 | 40.76 ± 18.34 |
| *Walker stand2run* | **265.50 ± 8.40** | 177.02 ± 62.48 | 182.71 ± 30.74 | 64.02 ± 31.54 |

*Table 1.* Comparison of few-shot transfer performance on eight test tasks in Cheetah and Walker. We report the mean and standard deviation of the cumulative reward at the end of training. Bold value indicates the best result.

agent can benefit from a joint world model for adapting to new individual tasks. Similarly, Nagabandi et al. (2019) propose a meta-learning approach where a dynamics model is trained to adapt quickly to new tasks by learning a prior over models. Hypernetwork-based methods (Huang et al., 2021) have been proposed to minimize forgetting while learning task-specific parameters in the multitask setting. Liu et al. (2024) introduces locality-sensitive sparse encoding to learn world models incrementally in a single task online setting. Kessler et al. (2023) investigate how different experience replay methods will affect the performance of MBRL. Related work for **model-based RL in general** can be found in Appendix B.

## 6. Conclusion

We proposed DRAGO, a novel approach for continual MBRL that effectively mitigates catastrophic forgetting and enhances the transfer of knowledge across sequential tasks. By integrating *Synthetic Experience Rehearsal* and *Regaining Memories Through Exploration*, DRAGO retains and consolidates knowledge from previous tasks without requiring access to past data, resulting in a progressively more complete world model. Our empirical evaluations demonstrate that DRAGO performs well in terms of knowledge retention and transferability, making it a promising solution for complex continual learning scenarios. Future work will explore extending DRAGO to larger-scale environments and more diverse task distributions.

## 7. Limitations

We only maintain one generative model throughout the continual training process, and this could potentially have mode collapse problem as the number of the tasks grows. The generative model is expected to capture the distribution of all prior tasks, which also relies on its own generated data. Thus the forgetting issue of the generative model will appear as its memory becomes "blurry" when the task number grows. To some extent, mixing the synthetic data with real world data will help mitigate this (note that the real world data can also come from the data collected by our reviewer,

which connects to the previous tasks), but the question of how we can better do continual learning for generative models remains and we leave it for future works. The current tasks tested in the paper are not highly complex, and there is a limited number of tasks, which can be the reason why we do not observe this problem in our setting. Developing continual generative models can be much more challenging, but also rewarding towards the goal of real continual agent.

## Impact Statement

This paper presents work whose goal is to advance the field of Machine Learning. There are many potential societal consequences of our work, none of which we feel must be specifically highlighted here.

## Acknowledgement

This work was supported by the Office of Naval Research (ONR) under grant #N00014-22-12592. Partial funding was also provided by The Robotics and AI Institute. The authors would like to thank David Abel, Lijing Yu, and Jingwen Zhang for their help and feedbacks on this project. This work was conducted using computational resources and services at the Center for Computation and Visualization, Brown University.

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

---

**Algorithm 1** DRAGO (Training process for each task)

---

**Require:** $\psi, \psi^-, \theta, \phi, \phi^-$: randomly initialized network parameters
     $T_{\text{old}}, E_{\text{old}}, G_{\text{old}}$: transition network and VAE up to the previous task
     $\eta, \tau, \lambda, \mathcal{B}^l, \mathcal{B}^r$: learning rate, coefficients, learner buffer, reviewer buffer

1: $T_\psi \leftarrow T_{\text{old}}$                  {load transition model from the previous task}
2: $E_\theta \leftarrow E_{\text{old}}$                   {load VAE encoder from the previous task}
3: $G_\theta \leftarrow G_{\text{old}}$                   {load VAE decoder from the previous task}
4: **while** not tired **do**
5:   // Collect episode with learner and reviewer models from $s_0 \sim p_0$:
6:   **for** step $t = 0, \ldots, \tau$ **do**
7:    $a_t \sim \Pi_\theta^l(\cdot | s_t)$                {Sample with learner model}
8:    $(s_{t+1}, r_t) \sim ENV(s_t, a_t)$               {Step environment}
9:    $\mathcal{B}^l \leftarrow \mathcal{B}^l \cup (s_t, a_t, r_t, s_{t+1})$            {Add to learner buffer}
10:   **end for**
11:   **for** step $t = 0, \ldots, \tau$ **do**
12:    $a_t \sim \Pi_\theta^r(\cdot | s_t)$                {Sample with reviewer model}
13:    $(s_{t+1}, \_) \sim ENV(s_t, a_t)$               {Step environment}
14:    $r_t = \text{calculate\_intrinsic\_reward}(s_t, a_t, s_{t+1})$        {Equation 7}
15:    $\mathcal{B}^r \leftarrow \mathcal{B}^r \cup (s_t, a_t, r_t, s_{t+1})$           {Add to reviewer buffer}
16:   **end for**
17:   update\_learner\_and\_reviewer$(\mathcal{B}^l, \mathcal{B}^r, \theta, \phi, \psi, \eta, \tau, \lambda)$       {Algorithm 2}
18:   update\_vae$(\theta, G_{\text{old}}, \mathcal{B}^l)$               {Algorithm 3}
19:   update\_transition\_from\_synthetic\_data$(\psi, T_{\text{old}}, G_{\text{old}})$       {Algorithm 4}
20: **end while**

---

# A. Algorithm Details

The DRAGO algorithm combines synthetic experience rehearsal and exploration-driven memory regaining to facilitate continual learning in model-based reinforcement learning (MBRL). This section provides a detailed, step-by-step breakdown of DRAGO, outlining how it maintains and updates both the dynamics and generative models throughout a sequence of tasks. For the first task, DRAGO exclusively trains the learner model and the rehearsal encoder-decoder pair using only online data.

### A.0.1. INITIALIZATION

For each task $\mathcal{T}_i$, DRAGO begins by randomly initializing the policy networks $\pi^{l,r}i$, the Q networks $Q_i^{l,r}$, and the reward networks $R_i^{l,r}$ for both the learner and reviewer models. These components are initialized separately, but they share a common transition network $T_i$.

The transition network $T_i$, along with the synthetic experience rehearsal encoder $E_i$ and decoder $G_i$, are initially randomly initialized for the first task. For subsequent tasks, these networks are loaded with the weights from the previous task's networks ($T_{i-1}, E_{i-1}$, and $D_{i-1}$). Notably, these previously trained components ($T_{i-1}, E_{i-1}$, and $D_{i-1}$) are employed as fixed modules for generating synthetic data, thereby supporting the rehearsal process without further updates.

### A.0.2. DATA COLLECTION

During each episode, both the learner agent and reviewer agent interact with the environment for the same number of time steps. The experiences $(s, a, s', r)$ encountered by each agent are stored in separate replay buffers: $\mathcal{B}_i^l$ for the learner and $\mathcal{B}_i^r$ for the reviewer. While the learner agent's rewards are directly sourced from the environment, the reviewer agent's intrinsic rewards are computed using the methodology outlined in Equation 7. This intrinsic reward mechanism drives the reviewer's exploration and memory regaining.

---

**Algorithm 2** update_learner_and_reviewer

---

**Require:** $\mathcal{B}^l, \mathcal{B}^r$: Learner and reviewer buffers

  $\psi, \psi^-, \theta, \phi, \phi^-$: Network parameters

  $\eta, \tau, \lambda$: Learning rate, coefficients

1: $\{s_t^l, a_t^l, r_t^l, s_{t+1}^l\}_{t:t+H} \sim \mathcal{B}^l$     {Sample trajectory from learner buffer}

2: $\{s_t^r, a_t^r, r_t^r, s_{t+1}^r\}_{t:t+H} \sim \mathcal{B}^r$     {Sample trajectory from reviewer buffer}

3: $r_{t:t+H}^{l'} \leftarrow$ calculate_reviewer_reward$(r_{t:t+H}^l)$

4: $J_\theta, J_\phi, J_\psi \leftarrow 0, 0, 0$     {Initialize loss accumulation}

5: $\hat{q}_1^l = Q_\theta^l(s_1^l, a_1^l)$

6: $\hat{q}_1^r = Q_\theta^r(s_1^r, a_1^r)$

7: $\hat{q}_1^{l'} = Q_\theta^r(s_1^l, a_1^l)$

8: $L_Q = \mathcal{L}_{\text{value}}(\hat{q}_1^l) + \mathcal{L}_{\text{value}}(\hat{q}_1^r) + \mathcal{L}_{\text{value}}(\hat{q}_1^{l'})$     {Calculate value loss at the first observation}

9: $\hat{r}_1^l = R_\phi^l(\hat{s}_1^l, a_1^l)$

10: $\hat{r}_1^r = R_\phi^r(\hat{s}_1^r, a_1^r)$

11: $\hat{r}_1^{l'} = R_\phi^r(\hat{s}_1^l, a_1^l)$

12: $L_R = \mathcal{L}_{\text{reward}}(\hat{r}_1^l) + \mathcal{L}_{\text{reward}}(\hat{r}_1^r) + \mathcal{L}_{\text{reward}}(\hat{r}_1^{l'})$     {Calculate reward loss at the first observation}

13: $J_\theta \leftarrow J_\theta + L_Q + L_R$     {Only update reward and value functions at the first step}

14: $\hat{s}_1^l = s_1^l, \hat{s}_1^r = s_1^r$     {Initialize the estimated first observations}

15: **for** $i = t, \ldots, t + H$ **do**

16:     $\hat{s}_{i+1}^l = t_\psi(\hat{s}_i^l, a_i^l)$

17:     $\hat{s}_{i+1}^r = t_\psi(\hat{s}_i^r, a_i^r)$

18:     $\hat{a}_i^l = \pi_\phi^l(\hat{s}_i^l, s_i^l)$

19:     $\hat{a}_i^r = \pi_\phi^r(\hat{s}_i^r, s_i^r)$

20:     $J_\phi \leftarrow J_\phi + \lambda^{i-t}(\mathcal{L}_\pi(\hat{a}_i^l) + \mathcal{L}_\pi(\hat{a}_i^r))$

21:     $J_\psi \leftarrow J_\psi + \lambda^{i-t}(\mathcal{L}_{\text{dynamics}}(\hat{s}_{i+1}^l) + \mathcal{L}_{\text{dynamics}}(\hat{s}_{i+1}^r))$

22: **end for**

23: $\phi \leftarrow \phi - \frac{\eta}{H}\nabla_\phi J_\phi$     {Update online network}

24: $\psi \leftarrow \psi - \frac{\eta}{H}\nabla_\psi J_\psi$     {Update online network}

25: $\phi^- \leftarrow (1-\tau)\phi^- + \tau\phi$     {Update target network}

26: $\psi^- \leftarrow (1-\tau)\psi^- + \tau\psi$     {Update target network}

---

**Algorithm 3** update_vae

---

**Require:** $\theta$: VAE parameters

  $G_{\text{old}}$: Previously trained VAE decoder

  $\mathcal{B}^l$: Replay buffer of the learner

1: $h \sim \mathcal{N}(0, 1)$

2: $(s^{\text{synth}}, a^{\text{synth}}) \leftarrow G_{\text{old}}(h)$     {Generate synthetic observations and actions}

3: $h^{\text{synth}} \leftarrow E_\theta(s^{\text{synth}}, a^{\text{synth}})$

4: $(\hat{s}^{\text{synth}}, \hat{a}^{\text{synth}}) \leftarrow G_\theta(h^{\text{synth}})$     {reconstruct synthetic observation and action}

5: $\{s_t^l, a_t^l, r_t^l, s_{t+1}^l\}_{t:t+H} \sim \mathcal{B}^l$

6: $h \leftarrow E_\theta(s_1^l, a_1^l)$

7: $(\hat{s}, \hat{a}) \leftarrow G_\theta(h)$     {reconstruct sampled observation and action}

8: $J_\theta = J_\theta + \mathcal{L}_{\text{gen}}(\hat{s}, \hat{a}) + \mathcal{L}_{\text{gen}}(\hat{s}^{\text{synth}}, \hat{a}^{\text{synth}})$

9: $\theta \leftarrow \theta - \frac{\eta}{H}\nabla_\theta J_\theta$     {update VAE parameters}

---

---

**Algorithm 4** update_transition_from_synthetic_data

---

**Require:** $\psi$: Transition network parameters

$\quad\quad\quad$ $T_{\text{old}}$: Previously trained transition network

$\quad\quad\quad$ $G_{\text{old}}$: Previously trained VAE decoder

1: $h \sim \mathcal{N}(0, 1)$

2: $(s^{\text{synth}}, a^{\text{synth}}) \leftarrow G^{\text{old}}(h)$ $\quad\quad\quad\quad\quad\quad\quad\quad\quad\quad\quad$ {Generate synthetic observations and actions}

3: $s' = T^{\text{old}}(s^{\text{synth}}, a^{\text{synth}})$ $\quad\quad\quad\quad\quad$ {generate next observation from old transition model}

4: $\hat{s'} = T_\psi(s^{\text{synth}}, a^{\text{synth}})$

5: $J_\psi \leftarrow J_\psi + \mathcal{L}_{\text{dynamics}}(\hat{s'}, s')$

6: $\psi \leftarrow \psi - \frac{\eta}{H} \nabla_\theta J_\psi$ $\quad\quad\quad\quad\quad\quad\quad\quad\quad\quad\quad\quad\quad\quad\quad\quad$ {Update online network}

---

### A.0.3. INFERENCE

The inference process in DRAGO is inspired by TD-MPC (Hansen et al., 2022), utilizing the Cross-Entropy Method (CEM) (Rubinstein, 1997) for action selection. During this process, a fixed number of trajectories of predetermined length are sampled and simulated using the current transition model $T_i$. For each trajectory, the cumulative return is calculated. The trajectories with the highest returns, referred to as elite trajectories, are selected to reshape the distribution of the initial actions. This iterative process is repeated for a fixed number of iterations, ultimately yielding a refined distribution over actions, which informs the final action selection. All the hyperparameters releated to the CEM algorithms is the same with TD-MPC (Hansen et al., 2022).

### A.0.4. UPDATING

DRAGO updates after each episode of rollouts for the same iterations as the number of rollout time-steps The updates tries to minimize the training objective, which is the sum of several losses wegihted temporally by a discount factor $\lambda$. Below is a detailed description of the loss functions used in the updates:

The transition model is updated using data from both the learner agent and the reviewer agent, as well as the synthetic observation-action pairs generated by the previous VAE decoder ($G_{i-1}$) and the subsequent observations generated by the previous transition model ($T_{i-1}$). This process maintains the transition model's accuracy for transitions encountered in previous tasks, thereby mitigating catastrophic forgetting of the world model. Given an observation $s$, an action $a$, and a target next state $s'$, the loss function calculates the mse between the predicted next observation using the transition model $T$ and the next state provided:

$$\mathcal{L}_{\text{dynamics}} = c_1 \|T_\psi(s, a) - s'\|_2^2$$

However, synthetic data updates for $T_i$ only occur at fixed intervals of steps to cope with the noise arising from inaccuracies in $G_{i-1}$ and $T_{i-1}$. This periodic updating strategy helps avoid noisy updates that can result from relying on outdated or inaccurate synthetic data.

Continual learning of the VAE ($E_i$ and $G_i$) occurs concurrently with the agent's updates. Data for this learning comes from both the state-action pairs obtained from the learner model's rollouts and the generated state-action pairs from $G_{i-1}$. The associated loss function for the VAE $\mathcal{L}_{gen}$ is shown in Equation 6.

The reward function $R$ which estimates the immediate reward from a given observation. The reward model enables the agent to estimate total return from a trajectory, and stabilizes the update for Q functions. It is updated using the following loss function:

$$\mathcal{L}_{\text{reward}} = c_2 \|R_\phi(z_i, a_i) - r_i\|_2^2$$

Additionally, the $Q$ functions for both agents are updated using the TD-objective shown as follows:

$$\mathcal{L}_{\text{value}} = c_3 \|Q_\phi(s_i, a_i) - (r_i + \gamma Q_{\phi^-}(s_{i+1}, \pi_\phi(s_{i+1})))\|_2^2$$

$Q$ and $R$ update only using the first steps of the horizons sampled, rather than using the complete horizon as in the original TD-MPC algorithm. This reduces the risk of noisy updates resulting from inaccuracies in the initial transition model.

The policy networks for both learner and reviewer agents are updated to maximize the expected $Q$ value using

$$\mathcal{L}_\pi = -Q_\phi(s, \pi_\phi(s))$$

In the above loss functions, $c_1, c_2, c_3$ are hyper parameters as weights for each losses.

## A.1. Hyperparameters

| Hyperparameter | Value (minigrid, cheetah, walker) |
|---|---|
| action repeat | 1, 4, 2 |
| discount factor | 0.99 |
| batch size | 512 |
| maximum steps | 100, 1000, 1000 |
| planning horizon | 10, (25, 15), 15 |
| policy fraction | 0.05 |
| temperature | 0.5 |
| momentum | 0.1 |
| reward loss coef | 0.5 |
| value coef | 0.1 |
| consistency loss coef | 2 |
| vae recon loss coef | 1 |
| vae kl loss coef | 0.02 |
| temporal loss discount ($\rho$) | 0.5 |
| learning rate | 1e-3 |
| sampling technique | PER(0.6, 0.4) |
| target networks update freq | 40, 2, 2 |
| temperature ($\tau$) | 0.01 |
| cost coef for reviewer reward ($\alpha$) | 0.5 |
| vae latent dim | 64, 256, 256 |
| vae encoding dim | 128 |
| mlp latent dim | 512 |
| gumble softmax temp | 1.0 |
| steps per synthetic data rehearsal | 10, 20 |

*Table 2.* Here we list the hyperparameters used for MiniGrid World, DM-Control cheetah, and DM-Control walker. Unlisted hyperparameters are all identical to the default parameters in TD-MPC.

## B. Related Work (Model-Based Reinforcement Learning)

Model-based reinforcement learning (MBRL) focuses on learning a predictive model of the environment's dynamics (Sutton, 1991). Learning world models (Ha & Schmidhuber, 2018; Hafner et al., 2019) specifically enables agents to accumulate knowledge about the environment's dynamics and generalize to new tasks or situations. By utilizing this model to simulate future states, agents can plan and make informed decisions without excessive real-world interactions. Most MBRL approaches can be categorized into two main categories in terms of how the learned model is used. The first category consists of methods that use the learned model to generate additional data and explicitly train a policy (Sutton, 1991; Pong et al., 2018; Ha & Schmidhuber, 2018; Sekar et al., 2020; Hafner et al., 2020; 2021; 2023), these approaches leverage the learned dynamics model to simulate experiences, which are then used to augment real data for policy optimization; the second category includes methods that learn the dynamics model and use it directly for planning to assign credit to actions (Ebert et al., 2018; Zhang et al., 2018; Janner et al., 2019; Hafner et al., 2019; Lowrey et al., 2019; Kaiser et al., 2020; Yu et al., 2020b; Schrittwieser et al., 2020; Nguyen et al., 2021; Zhang et al., 2024). These methods perform online planning by simulating future trajectories using the learned model to select actions without explicitly learning a policy. Recent approaches (Hansen et al., 2022; 2024) combine both techniques and achieves superior performance on various continuous control tasks. TD-MPC2 (Hansen et al., 2024) especially demonstrates the possibility of train a single world model on multiple tasks at once using MBRL.

## C. Tasks Specifications

Here we describe the specifications of the tasks included in this paper:

For MiniGrid World domain, all the tasks are to reach a goal. The pre-training tasks are dense-reward, and all fine-tuning tasks are sparse-reward.

- **Room1to2**: In this task we initialize the agent inside room 1 (top left, [11, 8]) and the goal inside room 2 (top right, [14, 9]).

- **Room1to3**: In this task we initialize the agent inside room 1 (top left, [8, 11]) and the goal inside room 3 (bottom left, [9, 14]).

- **Room3to4**: In this task we initialize the agent inside room 3 (bottom left, [11, 18]) and the goal inside room 4 (bottom right, [14, 17]).

For Deep Mind Control domain, all the pre-training tasks are from TD-MPC2 (Hansen et al., 2024), and the new fine-tune tasks are described below:

- **cheetah jump2run**: In this task we initialize the observation as a random state when the agent is performing the task "jump", then initialize the objective to be "cheetah run".

- **cheetah jump2back**: In this task, we initialize the observation as a random state when the agent is performing "jump", then initialize the objective to be "cheetah run backwards".

- **walker walk2run**: In this task, we initialize the observation as a random state when the agent is performing the task "walk", then initialize the objective to be "walker run".

- **walker run2back**: In this task, we initialize the observation as a random state when the agent is performing the task "run," then initialize the objective to be "walker run backwards".

- **walker back2run**: In this task, we initialize the observation as a random state when the agent is performing "run backwards", then initialize the objective to be "walker run".

- **walker stand2run**: In this task, we initialize the observation as a random state when the agent is performing the task "stand", then initialize the objective to be "walker run".

- **cheetah jump&run** In this tasks we encourage the agent to move forward in a high speed while their feet are both above the ground for a longer period of time. We averaged the rewards from cheetah run and cheetah jump with a lower threshold for speed and height.

- **cheetah jump&back** In this tasks we encourage the agent to move backwards in a high speed while their feet are both above the ground for a longer period of time. We averaged the rewards from cheetah run backwards and cheetah jump with a lower threshold for speed and height.

## D. Additional Results of Continual Training

We investigate whether the two components we proposed have side effect on the continual training tasks, where each two of them has relatively small overlap of transition dynamics and covers different state space. As shown in Table 3, DRAGO achieves similar performance with Continual TDMPC in all the training tasks, which is the MBRL baseline it is built upon, demonstrating that the proposed approaches will not deteriorate the training performance or induce more plasticity loss.

| Episode Reward | Cheetah run | Cheetah jump | Cheetah backward |
|---|---|---|---|
| DRAGO | 652.53 | 587.24 | 624.09 |
| Continual TDMPC | 675.31 | 646.30 | 580.59 |

| Walker run | Walker walk | Walker backward | Walker stand |
|---|---|---|---|
| 708.74 | 954.56 | 953.8 | 972.46 |
| 693.61 | 959.31 | 956.42 | 982.69 |

*Table 3.* Average Episode Return of the Continual **training** tasks after training for 1M steps.

## E. More Ablation Study Results

When trying the continual learning version for TDMPC, we find two interesting results. As shown in Figure 8 left, since we only transfer the dynamics model not the Q function, we thought excluding the Q value estimation in the planning process may yield better transfer results, but the result is the opposite. Without using the Q value in the planning process causes a performance drop. Moreover, in the original TDMPC implementation, a multi-step ahead prediction loss is used for updating the Q function and reward model, in the continual learning setting, we find that one-step prediction is better in complex environments like Deep Mind Control Suite as shown in the results of *Cheetah-jump*, which is the second one in Cheetah's continual training tasks.

We also investigate the influence of the frequency of *synthetic experience rehearsal*, the results are shown in Figure 8's second subfigure (from left to right).

In Figure 8's third subfigure, we show that if we also load Cheetah run's policy&value&reward, our method can reach even better results. However, this in practice requires prior knowledge that jump2run's reward function is similar to that of cheetah run. So it's not a scalable approach for now.

In Figure 8's last subfigure, we show a comparison of the effect of the planning horizon to the performance of DRAGO on Cheetah jump2back.

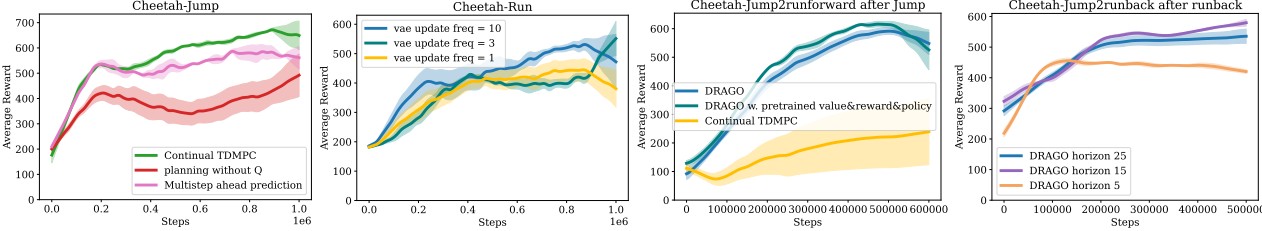

*Figure 8.* More ablation study results for continual TDMPC and DRAGO.

In Table 4, we compare with another baseline: Continual TDMPC + Curiosity, where we add the curiosity-based intrinsic reward to the continual TDMPC policy to increase exploration. We can see that DRAGO still outperforms this new continual MBRL baseline in all the four tasks. We should note that while this is a reasonable baseline, the comparison is a little unfair for our method as DRAGO can also be combined with any exploration method in a straightforward way. Specifically, while we have a separate reviewer model that aims to maximize our proposed intrinsic reward, our learner model that aims to solve each specific task can also be directly added with any intrinsic reward method like curiosity to encourage exploration, which does not contradict with the intrinsic reward of the separate reviewer.

We also try directly calculating the intrinsic reward of the reviewer and adding to the total reward of the learner, thus we do not need an additional reviewer model. As shown in Table 5, we see a large drop of performance for the continual training

| Average Reward | DRAGO | Curiosity + Continual TDMPC | EWC | Continual TDMPC | Scratch |
|---|---|---|---|---|---|
| *Cheetah jump2run* | **106.78 ± 32.01** | 88.36 ± 25.81 | 54.72 ± 62.72 | 93.96 ± 39.29 | 26.54 ± 2.67 |
| *Cheetah jump&run* | **248.92 ± 15.38** | 165.35 ± 67.01 | 156.98 ± 99.68 | 128.58 ± 100.14 | 182.77 ± 28.58 |
| *Cheetah jump2back* | **331.85 ± 11.05** | 133.81 ± 23.07 | 29.93 ± 7.15 | 73.98 ± 38.45 | 45.15 ± 4.92 |
| *Cheetah jump&back* | **147.30 ± 34.29** | 138.77 ± 45.55 | 117.92 ± 1.20 | 140.82 ± 28.00 | 129.75 ± 20.44 |

*Table 4.* Comparison of few-shot transfer performance on four test tasks in Cheetah. We report the mean and standard deviation of the cumulative reward at the end of training.

tasks, and this performance gap becomes larger and larger as the agent encounters more tasks, since it is encouraged to visit more and more possibly irrelevant states. Directly adding our intrinsic reward to the external reward and training only one single learner model makes it hard for the agent to complete the original task goal. If we only have one agent model (one policy), the intrinsic reward can have a side effect that 1. discourages the agent to visit places that it is already familiar with, thus hinders it to find the optimal solution to solve the task. 2. Encourages it to visit places that the previous mode is familiar with, which could be completely irrelevant for solving the current task. By having a separate reviewer policy that maximizes the intrinsic reward, we decouple the objectives. The learner policy focuses on maximizing the external reward to solve the current task effectively, while the reviewer policy explores states that help in retaining knowledge and connecting different regions of the state space. This separation allows both policies to operate without hindering each other's performance.

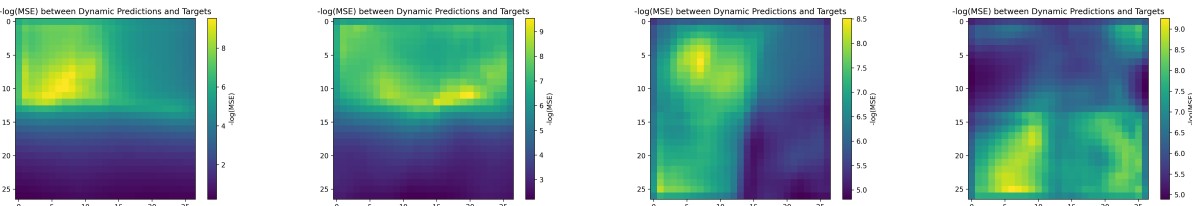

*Figure 9.* Prediction score heatmaps for **replay-based method** (**limited storage**). The heatmaps illustrates the accuracy of the dynamics model during continual learning by keeping a fixed part of the previous replay buffer (capped to a similar memory size as our generative model). Compared to DRAGO's performance in Figure 4, this method does not achieve the same qualitative result.

| Episode Reward | Cheetah run | Cheetah jump | Cheetah backward |
|---|---|---|---|
| DRAGO | 652.53 | 587.24 | 624.09 |
| DRAGO (Learner w. reviewer reward) | 583.13 | 403.70 | 330.73 |

*Table 5.* Average Episode Return of the Continual **training** tasks after training for 1M steps.

While in all our experiments above we evaluated DRAGO using TDMPC as the MBRL baseline, we also tried to combine DRAGO with another popular model-based RL baseline PETS (Chua et al., 2018), and show the preliminary results on the same MiniGrid tasks but with dense reward (we are not able to make PETS work on sparse reward settings unfortunately) in Table 7. DRAGO-PETS outperforms the baseline in 3 out of 4 tested tasks. Although we select finetune tasks that encourage the agent to perform in the union of sub observation spaces it has seen in previous tasks, we want to showcase the ability of DRAGO to retain knowledge by also comparing the few shot performance of pretrain tasks from only loading the world models of DRAGO and naive continual learning. Here, both world models are from a continual training of all four tasks in the Walker environment, in the order of Walker-Run, Walker-Walk, Walker-Stand, and Walker-Walk-Backwards.

| Average Reward (Dense) | DRAGO-PETS | Continual PETS |
|:---:|:---:|:---:|
| *MiniGrid1to3 after3* | **233.21 ± 21.07** | 150.84 ± 62.37 |
| *MiniGrid1to2 after2* | 101.03 ± 116.21 | **161.70 ± 44.31** |
| *MiniGrid1to3 after4* | **138.26 ± 99.05** | 43.04 ± 111.93 |
| *MiniGrid3to4 after4* | **234.65 ± 41.71** | 147.80 ± 105.84 |

*Table 6.* Comparison of few-shot transfer performance of PETS based methods on four test tasks in MiniGrid. We report the mean and standard deviation of the cumulative reward at the end of training.

| Average Reward | DRAGO | Naive TDMPC |
|:---:|:---:|:---:|
| *walker-run* | **371.46 ± 12.79** | 300.52 ± 27.07 |
| *walker-walk* | **740.89 ± 54.90** | 685.66 ± 43.05 |
| *walker-stand* | **871.17 ± 55.07** | 463.60 ± 107.12 |

*Table 7.* Comparison of few-shot performance of DRAGO and naive TDMPC continual learning by loading the world model at the end of pretraining on Walker tasks. We report the mean and standard deviation of the reward at 40k steps into the training.

