# OpenReview forum: "Knowledge Retention in Continual Model-Based Reinforcement Learning"
_ICML.cc/2025/Conference — ICML 2025 poster_

### Official Review · Reviewer_XZ8k · 2025-03-09

**Overall Recommendation:** 3

**Summary:**

The authors propose a method for continual model-based reinforcement learning, where, ideally, an agent retains previously learned skills while learning new skills mitigating the catastrophic forgetting problem. The main problem addressed is the bounded storage problem, where the agent is not assumed to have infinite storage for storing previous transitions, which most continual RL methods assume (replay-based approaches). Instead, this work consists of two key components:
1. Synthetic Experience Rehearsal, where a generative model (VAE) produces synthetic transitions from all previous tasks, which are used in tandem with the current tasks data to train the new transition model.
2. Regaining Memories Through Exploration, which is an intrinsic reward that encourages exploration of states that were well-understood in previous tasks but are not currently well understood. This supposedly helps to bridge newly learned tasks and previously learned tasks.

Hence, instead of storing previous transitions, the proposed method stores a generative model that learn the distribution of the previous transitions. Thus, the generative model should require less storage than keeping previous transitions.
The experimental evidence is conducted through continual learning experiments on the MiniGrid environment and the Deepmind Control Suite. The method is compared to the following baselines: TDMPC from scratch (model-based/world-model RL), 'continual' TDMPC (initialize with model of previous task), EWC (regularization-based continual RL).

The presented results significantly outperform the baselines in most experiments.

In an ablation study, the authors also verify the complementary roles of the two following components:
1. The generative model
2. The intrinsic reward

While I provided a "Reject" as score, I think that this paper overall details valuable research, that will be accepted if the authors improve the clarity and above all justify the use of the Generative Model to store experience through experimental support (comparing with CRL baselines).

**Claims And Evidence:**

The claims are not clearly stated, but I extracted the following.
- New tasks are learned without forgetting previous tasks.
	- Evidence: Supported by experiment on gridworld (nicely visualized in Figure 4)
- The previously learned skills improve few-shot transfer to new tasks that build upon the previous skills.
	- Evidence: supported by experiments on cheetah/walker
- Synthetic Experience Rehearsal learns the distribution of previous experiences
	- Evidence: partly shown in Figure4.
- The intrinsic reward incentivices agent to relearn previously learned tasks and bridges the gap
	- Evidence: only in ablation study that shows worse performance without it.
- Mitigating the bounded storage problem
	- Evidence: No evidence on environments with a large number of tasks, where replay-based continual RL would be limited.
		- Or alternatively: comparison to replay-based methods and how much less storage is needed for similar performance

**Essential References Not Discussed:**

See above

**Experimental Designs Or Analyses:**

The paper lacks experiments designed to test the performances on an environment with a larger number of tasks, which is especially relevant if the main goal is to be an alternative to replay-based methods in settings where they become infeasible due to a large number of previous transitions (bounded storage assumption). Could you extend the number of rooms to e.g. 16 ?
On a similar note, the method is not compared to replay-based methods, which would probably perform better or similar on the conducted experiments (according to my CL expert fellows).

Maybe a valuable experimental question would thus be: "how close can we match their performance with x less storage use?"

TD-MPC baselines: Please clarify how exactly 'from scratch' and continual TDMPC differ.
Does the continual TDMPC use the same task-embedding as the model from the previous task?

Some baselines are missing. For example: pseudo-rehearsal (Atkinson et al.) and world-model pseudo-rehearsal (Ketz et al.) -> see below.

**Methods And Evaluation Criteria:**

The proposed methods and evaluation criteria make sense for the problem at hand.

However, the evaluation would benefit from another experiment demonstrating good performance where replay-based methods fail to verify the main claim of solving the bounded storage problem. Specifically, as the authors are reusing the technique of learning a generative model instead of storing the raw replay buffer, they should demonstrate the extent to which this technique is more efficient than storing a replay buffer. Thus, the authors could use replay buffers with limited capacity (to match the storage requirements of their generative model). Then, uniformly drawing could constitute a first baseline, even if more advance storage selection and sampling have been developed in CL papers.

**Other Comments Or Suggestions:**

Background formalism MDP tuple: Usually use <> instead of ()
Typos:
- Chapter title 3
- Lines 4-7 under eq. 5
- Just before chapter 4: 2x same sentence.

While I graded the paper with a reject. I think that the paper does provide a valuable research work, that, once improved, will be of valuable interest for many readers. However, in its current form, it cannot be accepted to a major conference such as ICML. I hope that the authors can use the provided feedback to improve their work.

**Other Strengths And Weaknesses:**

Strength:
- Figure 4 is really nice, although a short claim as first sentence what this figure shows would be great.
In general you could improve the captions of all figure following this principle:
   - The first sentence should highlight the main message of the Figure/Table. E.g. **DRAGO reduce the catastrophic forgetting from previous tasks,**
   - The next sentences then explain what is depicted in the Table/Figure, E.g Illustrated by the prediction score of the learned world models across the entire gridworld after each task.... etc

Weaknesses:
- Missing clear description of all research questions answered / main contributions.
- Figures:
	- architecture overview
	- do not always contain a direct description of what to take away (as explained above)
	- Figure 3 does not add a lot -> remove?
	- Figures 5,6 could use a grid
	- missing number of seeds/runs (also at table 1)
- Formalism:
	- TDMPC2 uses the MSE on the embedded state. Are the states embedded in DRAGO?
		- If not: This is not really TDMPC -> clarify in method
		- If yes: not clear from context, redefine s.
	- The transition model $T_i$ and it's probability are never properly defined. What kind of model is it?
	- Makes it sound like its two different models (also with eq.4 and 5 using one each)
	- Suggestion: Just use $p_T(s'|s,a;\psi)$ and $\mathbb{E}[p_T(s'|s,a;\psi)]$ , or define T that way.
	- Loss formulation is confusing, because you show the total loss using probabilities and then split it later into dyn and gen loss, whereas the dyn loss is actually MSE that uses the deterministic(?) T.
	- Also, the total loss (eq. 4) does not specify, where $\hat{s}'$ comes from, which is only later revealed to be from $T_\text{old}$ .
		- Maybe equations 3 and 4 are unnecessary (and confusing) and can be ommited altogether?
		- Instead just show L_dyn and L_gen and then: L_total = L_dyn + L_gen
- Baselines:
	- Continual TDMPC is not really continual, more like 'pre-initialized'
	- Missing comparison to a replay-based method to show sota results and how drago compares.
- Limitations missing in main paper. Super important, since the generated synthetic state-action pairs may not fully represent all previous tasks with a growing number of tasks, as discussed in Appendix F.
- A (short) high-level algorithm in the main paper would also be beneficial.

More specifically, to improve clarity, I strongly advise the authors to include in their paper:
I. A list of contributions at the end of the introduction (often denoted with i., ii., ...) that stands out visually.
II. A list of scientific questions at the beginning of the experimental evaluation section (often denoted Q1, Q2, ... etc), that are each answered in different paragraphs. For example
Q1/ How does DRAGO perform in comparison to existing CRL baselines?
Q2/ How much does the generative model help preventing catastrophic forgetting?
Q3/ Is the generative model more storage efficient than the classical use of a replay buffer?
...

**Questions For Authors:**

- Q1: add research questions / key contributions
- Q2: properly define transition model/likelihood, also focus on one notation (T or p)
- Q3: reformulate loss definitions for clarity (see above)
- Q4: add limitations + high-level algorithm
- Q5: add replay-based baseline -> how does DRAGO improve upon that? (e.g. how much % less storage needed for similar performance))
- Q6: add pseudo-rehearsal baselines (see references above)
- Q7: benchmark number of tasks -> when does it fail?

**Relation To Broader Scientific Literature:**

This research work combines known methods from continual learning (pseudo-recursal) with methods from reinforcement learning (MBRL).
Some essential references are not discussed:
- Pseudo-recursal is not referenced, neither is pseudo-rehearsal.
	- which is exactly the idea of [1], just with DQN instead of MBRL
There even exists methods applying pseudo-rehearsal to MBRL [2] (but not published)

How does your method differ? You should explain why these baselines are excluded, or if they are not relevant to support your claims (but then the reader would need explicit claims listed)

[1] Atkinson, Craig, et al. "Pseudo-rehearsal: Achieving deep reinforcement learning without catastrophic forgetting." *Neurocomputing* 428 (2021): 291-307.
[2] Ketz, N., Kolouri, S., & Pilly, P. (2019). Continual learning using world models for pseudo-rehearsal. arXiv preprint arXiv:1903.02647.

**Theoretical Claims:**

Not applicable

---

> ### Author Rebuttal · Authors · 2025-04-01
>
> Thank you for the detailed and constructive feedback. We are encouraged that you find our ideas valuable and appreciate your suggestions on clarity and comparisons. We address each concern below:
>
> Q: Replay-based Baseline
>
> A: Thank you for pointing this out – we are adding a **fixed-size replay buffer** baseline with the same storage budget as DRAGO’s generative model. Preliminary implementation is underway, and we will share results on this anonymous link https://drive.google.com/file/d/18TdI9nPCT7MMQCQMmL1ZYxJkblhUI91i/view?usp=sharing after we have the results. This baseline is an experience replay approach that stores a limited number of real past transitions (capped to a similar memory size as our generative model) and interleaves them during training on new tasks. Our hypothesis is that DRAGO will perform on par with or better than this bounded replay baseline, especially as tasks accumulate (since DRAGO can simulate diverse past experiences without strictly limited slots).
>
> Q: Pseudo-rehearsal Baseline
>
> A: Since the original paper does not provide source code, we are currently in the process of implementing a **world-model pseudo-rehearsal** baseline based on the paper. This is a generative replay baseline where an agent uses a pretrained world model (or VAE) to rehearse past tasks’ experiences, akin to DRAGO’s Synthetic Experience Rehearsal but without our intrinsic exploration component and continual learning of the generative model. We will share results on this anonymous link https://drive.google.com/file/d/18TdI9nPCT7MMQCQMmL1ZYxJkblhUI91i/view once the experiments are finished.
>
> Q: Terminology “Continual TDMPC”:
>
> A: Thank you for flagging this potential misunderstanding. In our paper, “Continual TDMPC” refers to the naïve baseline where we initialize the model for each new task with the previously learned TDMPC world model, without any forgetting mitigation. It is essentially the standard TDMPC algorithm simply fine-tuned sequentially.
>
> Q: Paper structure and readability:
>
> A: We appreciate these suggestions and will incorporate them in the final version. We will add a bulleted summary of contributions in the introduction. Specifically,
>
> - **Novel Continual MBRL Framework**. We introduce **DRAGO**, a new approach for continual model-based reinforcement learning that addresses catastrophic forgetting while incrementally learning a world model across sequential tasks without retaining any past data.
>
> - **Synthetic Experience Rehearsal**. We propose a generative replay mechanism that synthesizes “old” transitions using a learned generative model alongside a frozen copy of the previously trained world model. This synthetic data consistently reinforces earlier dynamics knowledge, mitigating forgetting in each new task.
>
> - **Regaining Memories Through Exploration**. We design an intrinsic reward signal that nudges the agent toward revisiting states that the old model explained well—effectively “reconnecting” current experiences with previously learned transitions. This mechanism complements the synthetic rehearsal by incorporating real environmental interactions to maintain a more complete world model.
>
> - **Extensive Empirical Validation**. Through experiments on **MiniGrid** and **DeepMind Control Suite** domains, we show that DRAGO:
>
> 1. Substantially improves knowledge retention compared to standard continual MBRL baselines.
>
> 2. Achieves higher forward-transfer performance, allowing faster adaptation to entirely new (but related) tasks.
>
> 3. Exhibits strong few-shot learning capabilities, substantially outperforming both learning-from-scratch and other continual methods under limited interaction budgets.
>
> Q: Formalism clarity (transition model, loss equations, notation)
>
> A: We apologize for the confusion. In our notation, $T$ is the parametric transition model (the learned dynamics predictor) and $p(\cdot)$ denotes a probability or distribution. For example, $p(s' \mid s,a;\psi)$ is the likelihood of observing $s'$ given $(s,a)$ under the transition model $T_{\psi}$. We will explicitly define the transition model in the main text upon first use and use a consistent notation (e.g. using $T$ for the model and $p$ for probabilities) throughout. We will also clarify the loss equations step-by-step. In particular, Equation (4) in the paper decomposes the loss into current-task loss (on real data $D_i$) and synthetic rehearsal loss (on generated data $\hat D$). Equation (5) then shows the combined training objective for the dynamics model: it includes a term $|T_i(s,a)-s'|^2$ for real transitions and a term $|T_{i}(ŝ,â)-T_{old}(ŝ,â)|^2$ for synthetic ones (weighted by $\lambda$). We will make sure to clearly explain each term and the roles of $T$ vs. $p$ in a revision.
>
> Q: Minor issues (typos, notation, algorithm pseudocode)
>
> A:  We will fix all minor typos and notational errors. We also agree that a high-level pseudocode algorithm in the main paper would aid clarity.

---

> > ### Comment · Reviewer_XZ8k · 2025-04-03
> >
> > While I believe most of my claims have been adressed, I still have 2 points I that I would like the authors to clarify. Could you comment exactly what makes DRAGO better than the 2 baselines in your paper?
> > Particularly, I find the second plot very weird, why does the replay-based baseline's performances complitely drop ?
> >
> > Also, I would recommend to drop *extensive* for this evaluation. While I believe that if the authors extend their evaluation of the two additional baselines, it would be sufficient to support their claims, I don't consider it to be *extensive* (but I might be wrong).

---

> > > ### Author Response · Authors · 2025-04-03
> > >
> > > Thank you for the follow-up questions.
> > >
> > > - Why DRAGO Is Better Than the Replay-Based and Pseudo-Rehearsal Baselines
> > >
> > > Bounded Replay vs. Generative Replay: The replay-based baseline can only store a small fraction of previous transitions in memory, so as the number of tasks grows, past data coverage shrinks. In contrast, DRAGO’s generative model synthesizes essentially unlimited “old” transitions, preserving a broader variety of past experiences.
> > >
> > > Reviewer Reward for Exploration: Our intrinsic “reviewer” reward actively drives the agent to revisit states that connect the new task with old tasks, resulting in a more unified, accurate world model. The bounded-replay baseline cannot exploit a similar “exploratory bridge,” so it remains siloed in the new task data plus a few replay samples.
> > >
> > > For the pseudo-rehearsal baseline, since it pretrains a variational autoencoder on early, randomly collected rollouts, it's hard for it to cover diverse transitions for both old and new tasks. Additionally, it also does not have our reviewer reward which helps the agent connect the transitions from current task and the old tasks.
> > >
> > > - Why the Replay Baseline’s Performance Drops in the Second Plot
> > >
> > > Our hypothesis is that the world-model for the replay-based agent is highly imprecise early on—so it occasionally “lucks into” hitting the goal (giving a temporary spike). As learning proceeds without a sufficiently diverse buffer of old data, it re-overfits to the most recent task or gets stuck, causing the subsequent performance drop. Note that, despite the transient fluctuations, its highest average reward remains low (around 10 in the plot), whereas the other two methods compared reach 60 and 100.. Thus the “drop” is exaggerated on a small scale—this method does not truly master the tasks but rather hovers around a low-performance regime.
> > >
> > > We used the word *extensive* to describe all the experiments we included in the paper - that answer was meant to be the list of contributions we plan to include in the introduction section based on the reviewer's suggestion.

---

### Official Review · Reviewer_rfpF · 2025-03-11

**Overall Recommendation:** 3

**Summary:**

This paper presents a new approach to model-based continuous reinforcement learning (DRAGO) aimed at improving the incremental development of world models across a range of tasks. DRAGO consists of two key components: Synthetic Experience Rehearsal and Regaining Memories Through Exploration. Empirical evaluations have shown that DRAGO is capable of preserving knowledge across a wide range of tasks and achieving excellent performance in a variety of continuous learning scenarios.

## update after rebuttal
I have read the authors' rebuttal and appreciate their hard work and detailed responses. While their clarifications have helped me better understand certain points that were previously unclear, they are not sufficient for me to increase my rating score further.

**Claims And Evidence:**

YES

**Essential References Not Discussed:**

No significant relevant papers were found that were not cited.

**Experimental Designs Or Analyses:**

The experimental design is generally sound, but still insufficient, and additional supplementary experiments need to be added.

**Methods And Evaluation Criteria:**

YES

**Other Comments Or Suggestions:**

N/A.

**Other Strengths And Weaknesses:**

Strengths:
1. The paper explains the innovativeness clearly enough, and the structure of the article and the linguistic description make it very easy for the reader to understand the proposed innovativeness.
2. The presentation of previous relevant work is clearer and may be a continuation of the team's previous work.
3. The algorithms and pseudocode are more adequately described and innovatively explained in the supplementary material.
4. The introduction of the problem through the robot was very easy to understand

Weaknesses:
1. In Section 3.1, the concept of “Synthetic Experience Rehearsal” is introduced as a method for generating synthetic experiences from past tasks. Could the authors clarify how the synthetic experiences generated by the generative model differ from actual past experiences in terms of their representational accuracy?
2. Too few algorithms for comparative testing. It is recommended that the authors add more SOTA methods on continuous reinforcement learning (lifelong reinforcement learning) published in top journals/conferences in recent years
3. In Eq. (1), where the synthetic state is generated using the frozen old world model Told, could the authors provide more explanation about how the state-action pair is sampled from the generative model p_G(s, a; \theta)? Specifically, how is the action sampled in continuous action spaces, and how do the learned transition dynamics handle this?
4. It’s suggested to quantitatively demonstrate the degree of forgetting in the main text, e.g., by reporting how well each method recovers performance (or world model error) on earlier tasks after the training sequence has ended. While Table 3 in the Appendix of the paper provides a comparison of the final performance of DRAGO and the baseline on each of the training tasks, it would have been more convincing in the main paper to mention that ‘DRAGO roughly maintains the performance of the old tasks without degrading the performance of the new tasks’.
5. On some of the test tasks, the ‘train from scratch’ approach came close to or even outperformed the continuous learning baseline (for the ‘Cheetah backward’ related combinatorial task). It is suggested that the authors briefly analyse the reasons for this phenomenon.
6. There still seems to be a clerical error in the title and abbreviation.

**Questions For Authors:**

N/A.

**Relation To Broader Scientific Literature:**

The task addressed in this paper is a hot topic of discussion in recent years, where models learn new tasks with catastrophic forgetting of past tasks. This has been a great challenge in the field of continuous learning and reinforcement learning, and there have been many previous studies on continuous reinforcement learning (lifelong reinforcement learning). The authors' main innovation comes from a research finding on dreaming in the 1990s, which has been applied to other modelling domains, but not to the field of continuous reinforcement learning (lifelong reinforcement learning).

**Theoretical Claims:**

I examined the theoretical proofs of two key components of the author's approach: Synthetic Experience Rehearsal and Regaining Memories Through Exploration. The logic is basically correct, but there is still a lack of clarity that needs to be improved.

---

> ### Author Rebuttal · Authors · 2025-04-01
>
> We thank the reviewer for their positive feedback and thoughtful questions.We address your questions below.
>
> Q: “how the synthetic experiences generated by the generative model differ from actual past experiences in terms of their representational accuracy?”
>
> A: Synthetic experiences in DRAGO are generated by a continually learned VAE-based generative model $G$ that **encodes and decodes both states and actions**, capturing the joint distribution of prior state-action. In other words, the agent “dreams” trajectories from its learned world model. These synthetic experiences **approximate real past interactions**—if $G$ is well-trained, the sampled states and actions resemble those from earlier tasks, helping reinforce previously learned dynamics **without storing actual data**. We acknowledge that synthetic data may not be perfect (due to model approximation error, we also discussed this in appendix F), but our design mitigates this: we **retrain $G$ after each task on a mix of new and generated old data** so it retains the ability to produce samples representative of all past tasks. Moreover, **DRAGO’s second component (Regaining Memories via Exploration)** complements generative rehearsal by actively revisiting important states in the real environment. This intrinsic reward-driven exploration addresses any gaps in the generative model’s coverage, ensuring that the world model doesn’t diverge from true environment dynamics.
>
>
> Q: Additional Continual Reinforcement Learning Baseline
>
> A: We agree that incorporating more recent state-of-the-art baselines will strengthen the evaluation. In the original submission, we compared against **naïve continual fine-tuning (Continual TDMPC)**, a **from-scratch retraining baseline**, and **EWC**. To address the reviewer’s suggestion, we are **running two new baselines** and will include them on this link https://drive.google.com/file/d/18TdI9nPCT7MMQCQMmL1ZYxJkblhUI91i/view?usp=sharing once we get the results: **(1) Pseudo-rehearsal with world models**: a generative replay baseline where an agent uses a learned world model (or VAE) to rehearse past tasks’ experiences, akin to DRAGO’s Synthetic Experience Rehearsal but without our intrinsic exploration component and continual learning of the generative model. **(2) Bounded replay-buffer baseline: an experience replay approach that stores a limited number of real past transitions (capped to a similar memory size as our generative model)** and interleaves them during training on new tasks. This will provide a direct yardstick for DRAGO’s no-data approach: i.e., how well does compressing experiences into a model compare to simply saving raw data with equal storage budget.
>
> Q: “How is the action sampled in continuous action space for synthetic data”
>
> A:  We sample actions jointly with states from the generative model, rather than picking random actions. This is crucial for continuous action domains: **random actions might not lead to meaningful or realistic transitions**, whereas our generative model produces plausible $(s, a)$ pairs grounded in past experience​.
>
> Q: **Details on Sampling from $p_G(s,a;\theta)$ in Eq. (1)**
>
> A: In Equation (1) of the paper, $(\hat{s}, \hat{a}) \sim p_G(s,a;\theta)$ denotes drawing a state-action pair from the generative model’s distribution. As explained above, this is implemented by sampling from the VAE. We will clarify in the text that $p_G$ is the VAE-generative distribution over state-action pairs. The sampled pair $(\hat{s}, \hat{a})$ is then fed into the **frozen previous dynamics model $T_{\text{old}}$** to predict the next state $\hat{s}' = T_{\text{old}}(\hat{s}, \hat{a})$. This yields a synthetic transition $(\hat{s}, \hat{a}, \hat{s}')$ which is used (alongside real data from the new task) to train the current dynamics model. Thus, **the generative model provides realistic past states and actions, and $T_{\text{old}}$ ensures the next-state is generated according to learned physics**.
>
> Q: Train from scratch approach’s performance
>
> A: You are correct that in a few scenarios (notably the Cheetah run-backward task), the scratch baseline slightly overtakes continual learning baseline. We observed this in our results: the continual learning method does **not fully eliminate plasticity loss**, and a sufficiently different new task can benefit from a fresh model. We hypothesize that in the backward-running tasks, the agent’s prior knowledge (largely acquired from forward-running dynamics) was less applicable or even somewhat **biasing the exploration**. We thank the reviewer for pointing this out, and we will clarify in the revision.

---

### Official Review · Reviewer_dsES · 2025-03-13

**Overall Recommendation:** 4

**Summary:**

The work aims to develop a new model-based reinforcement learning method that trained on a set of tasks with consistent changes and different reward functions. The researchers assume that the environment's dynamics remain the same for all tasks. They use TD-MPC as the basic approach, and train a separate generative model (VAE) that simulates the generalized transition function using a dataset of the new task and synthetic data from the old version of the model. The current transition model is then trained using an error minimization loss function on the current dataset and error regularization on the old model's on data generated by the generative model. Additionally, the authors propose using internal rewards to encourage visits to states that are accurately predicted by both the old and current models.
To evaluate the effectiveness of their approach, the authors compare it with traditional TD-MPC and an older EWC baseline on two tasks: miniGrid and MuJoCo.

## update after rebuttal
Both before and after the rebuttal phase, I believe that the work has a certain novelty, and therefore, I leave my current high assessment.

**Claims And Evidence:**

The main claims of the authors regarding the effectiveness of using the generative model for previous tasks are confirmed by experiments.

**Essential References Not Discussed:**

The authors have provided all the necessary references and methods.

**Experimental Designs Or Analyses:**

It should be noted that the test and training tasks are not randomized, and the authors perform experiments on fixed sequences of tasks. This raises questions about the generalizability of their results to other sequences and compositions of test tasks. Additionally, the authors use an old EWC continuous learning baseline, which is a significant drawback and does not reflect the current state of the field.

**Methods And Evaluation Criteria:**

The method itself is relatively new, and while it mainly utilizes the TD-MPC code base, it does provide an improvement in the continuous learning process. I should note that TD-MPC also claims to have some form of multitasking ability and the capacity to work with multiple tasks at once. However, the authors do not elaborate on this in any detail. Instead, the authors use relatively simple vector control systems to create a series of tasks that need to be solved. The method in question is limited to vector envs only and is unlikely to be effective for observations presented in the form of images.

**Other Comments Or Suggestions:**

The authors did not decipher the notation q_\theta_i in formula 6.

**Other Strengths And Weaknesses:**

It would also be useful to compare the authors' approach with different curriculum learning scenarios based on the same sequence of tasks. It would also be a good baseline.

**Questions For Authors:**

To what extent would any curriculum learning scenarios also be effective for such task sequences?

**Relation To Broader Scientific Literature:**

The authors did not adapt or demonstrate the effectiveness of other continuous learning methods mentioned in the review in any way. Yes, the environment model is not used there, but perhaps this is not so necessary for such simple environments as Minigrid and MuJoCo.

**Theoretical Claims:**

The paper provides a conclusion of the main loss function in equation 5, which can be considered sufficient theoretical justification for the proposed method.

---

> ### Author Rebuttal · Authors · 2025-04-01
>
> Thank you for your very positive evaluation and accept recommendation. We believe we can resolve your concerns.
>
> Q: TDMPC2’s Multitask Training
>
> A:We clarify that TDMPC2 is evaluated in a multitask regime: it is trained on all tasks jointly, with access to the full replay buffers from every task simultaneously and (implicitly) the task identity or reward function for each experience. This means TDMPC2 benefits from seeing all task data at once (no task ordering) and can leverage task-specific information during training. In contrast, **DRAGO** tackles tasks in a strict continual learning manner: tasks are presented sequentially, and our method does not utilize task IDs or any access to past task data/replay buffers once those tasks are finished. DRAGO must retain knowledge through its mechanisms (generative rehearsal and intrinsic reward) without being able to directly revisit past data.
>
> Q: Additional Continual Learning Baselines
>
> A:We agree that incorporating more recent state-of-the-art baselines will strengthen the evaluation. In the original submission, we compared against naïve continual fine-tuning (Continual TDMPC), a from-scratch retraining baseline, and EWC. To address the reviewer’s suggestion, we are **running two new baselines** and will include them on this link https://drive.google.com/file/d/18TdI9nPCT7MMQCQMmL1ZYxJkblhUI91i/view?usp=sharing once we get the results: **(1) Pseudo-rehearsal with world models**: a generative replay baseline where an agent uses a fixed pretrained world model (or VAE) to rehearse past experiences, akin to DRAGO’s Synthetic Experience Rehearsal but without our intrinsic exploration component. **(2) Bounded replay-buffer baseline**: an experience replay approach that stores a limited number of real past transitions (capped to a similar memory size as our generative model) and interleaves them during training on new tasks. This will provide a direct yardstick for DRAGO’s no-data approach: i.e., how well does compressing experiences into a model compare to simply saving raw data with equal storage budget.
>
> Q: Curriculum Learning
>
> A:Thank you for the interesting suggestion on curriculum learning. We agree that intelligently ordering tasks (e.g., from easier to harder or with gradually increasing complexity) could further improve continual learning performance. A curriculum might help the agent build up its world model in a more structured way, potentially reducing forgetting and improving transfer. However, designing an optimal curriculum is non-trivial and was beyond the scope of our current work, which focuses on general mechanisms applicable to any task sequence. We opted to evaluate DRAGO on diverse task sequences without assuming a favorable order. Nonetheless, exploring curriculum learning in conjunction with DRAGO is an exciting avenue for future work. We will note this in the discussion as a potential enhancement, as it could complement our approach by easing the learning progression through tasks.
>
> Q: Handling Image Observations
>
> A: Thank you for the suggestion. We would like to first point out that the domains that we tested on (MIniGrid & Deepmind Control Suite) are two of the most popular RL benchmarks that have been evaluated in a large number of prior papers and have been shown to be quite challenging environments. The tasks we designed for evaluating transfer performance are even more challenging on DMC tasks as they require the agent to learn to transition from one locomotion mode (jump, run etc.) to another (run forward, run backward). While due to time constraints we cannot test the method on image observation settings, we think this is an exciting and challenging problem for future work as we mentioned at the end of Section 3.1, especially as we can replace VAE with diffusion models that are capable of generating high-quality image data.
>
> Q: Clarification of Notation $q_{\theta_i}$ in Eq. (6)
>
> A: We apologize for the confusion regarding the notation $q_{\theta_i}$ in Equation (6). In the context of our VAE-based generative model, $q_{\theta_i}$ denotes the encoder’s approximate posterior distribution for task $i$. In other words, $q_{\theta_i}(z \mid s,a)$ is the VAE encoder’s output: the probability distribution (in latent space) that approximates the true posterior of the latent variable $z$ given an observation-action pair $(s,a)$. The subscript $i$ indicates that this encoder (with parameters $\theta_i$) is the one learned up to task $i$ (since we train a new generative model $G_i$ for each task $i$ using both new and past data via rehearsal). We will explicitly clarify this in the revised paper. Essentially, Eq. (6) is the standard VAE loss: $L_{\text{gen}}(\theta_i) = E_{(s,a) \sim D_{\text{gen}}}\Big[ -E_{z \sim q_{\theta_i}(z|s,a)}[\log p_{\theta_i}(s,a \mid z)] + \text{KL}(q_{\theta_i}(z|s,a)|p(z)) \Big]$, where $q_{\theta_i}$ is the encoder’s distribution and $p_{\theta_i}$ is the decoder (generative distribution).

---

### Official Review · Reviewer_5Rrm · 2025-03-30

**Overall Recommendation:** 3

**Summary:**

The authors introduce a new method (DRAGO) aimed at mitigating catastrophic forgetting in model-based RL in situations where previous experience cannot be stored. The authors propose to learn world model that compresses experience of previous tasks, and propose a novel intrinsic reward which encourages the policy to bridge the gap between different tasks, towards learning a more complete world model. The authors test their method empirically in various continual RL gridworld and continuous control domains, showing improvement over vanilla MBRL baselines and a previously successful continual-learning method (EWC).

**Claims And Evidence:**

The motivation lacks justification. For instance, are storage limitations really a bottleneck for policy learning in the present day? Can the authors qualify this with quantitative estimates? Similarly, do they have specific examples of privacy-preservation preventing the training of a generalist policy? The on-device deployment angle seems to most well justify the need for memory. However, there the justification suffers from another problem. Do the authors believe that online MBRL is likely to take place on-device, as opposed to in-context learning of some appropriately meta-trained foundation model (as is becoming standard for many AI applications)?

The results in Figures 4 and 5 are convincing that this method improves over the chosen baselines. The authors test their method on both gridworlds and continuous control tasks, and in both settings they see improvement. The ablation study in Figure 6 is also convincing. It would be useful for the authors to include information on the number of seeds they ran for each method in the figure captions.

As far as I can tell, the choice of continual RL tasks is not standard, and was rather determined by the authors. This raises the concern that these tasks could have been cherry-picked specifically to showcase the benefit of this method. Can the authors comment on why they did not use pre-defined standard benchmarks here, and whether there have any additional data or arguments that could allay a reader's concern that these tasks have been deliberately chosen to fit the specific setup in which the method is likely to succeed?

**Essential References Not Discussed:**

N/A.

**Experimental Designs Or Analyses:**

The evaluation setup is insufficiently well described. Is RL taking place during the "test" tasks or are these purely in-context learning? I assume that RL is still taking place, but then the designation of these as "test" tasks seems rather odd (usually one things of a train-test split, where there is no in-weights learning taking place on the test split). More explanation is required to convince me that the evaluation procedure is reasonable, rigorous and reflects a likely real-world deployment setup as motivated in the introduction.

**Methods And Evaluation Criteria:**

The DRAGO algorithm is well-described, and I believe that there are sufficiently many details provided that this work would be reproducible. The loss functions in equations (5) and (6) appear to be correct to me. The intrinsic reward in equation (7) is well-motivated, if a little ad-hoc. It would be useful to have a system diagram summarizing the various components of the full system (VAE, dynamics model, intrinsic reward).

The integration with TD-MPC is rather confusingly described to me. As I understand it, the philosophy of TD-MPC is to be encoder-only. However, here the authors specify that they need a decoder for state prediction. Can the authors comment on why they did not stick with the encoder-only philosophy of TD-MPC. It is also unclear to me how the "learner" and "reviewer" portions of the agent are combined when using MPC for planning. Can the authors comment on why they did not simply learn one value function on the sum of the intrinsic and extrinsic reward, and how the reviewer and learner combine to produce the policy?

This sentence is also confusing to me: For each new test task, we randomly initialize the reward, policy and value models and reuse only
the world model (dynamics)". Why are the reward, policy and value models thrown away for test tasks if these are kept throughout training? Is this because of an empirical performance gain or because it would be unfair to keep them around for test, for some reason. More justification is needed here.

**Other Comments Or Suggestions:**

N/A.

**Other Strengths And Weaknesses:**

N/A.

**Questions For Authors:**

Please see the questions in my responses above.

**Relation To Broader Scientific Literature:**

Both the MBRL and continual RL literatures are well-reviewed. One line of work that it may be useful to additionally mention in this regard is world models learned from data (e.g. Genie https://arxiv.org/pdf/2402.15391, UniSim https://arxiv.org/abs/2310.06114). Much as in the same way that large language models meta-learning on internet scale data has led to foundational models capable of long in-context continual adaptation, the same may be true of world models. It would be interesting to hear the authors thoughts on this complementary direction and how their work could fit into that narrative, were it to transpire to be successful at scale.

**Theoretical Claims:**

N/A

---

> ### Author Rebuttal · Authors · 2025-04-01
>
> Thank you for the thoughtful review and for acknowledging the strong empirical results of DRAGO, as well as the clarity of our writing. We address your concerns below:
>
>
> Q: bounded memory and On-device MBRL
>
>
>
>
> A: We agree that **bounded-memory continual learning** is most critical in constrained or privacy-sensitive scenarios. In practice, real agents **cannot always store unlimited replay data**. For example, (i) *on-device learning*: robots or mobile devices have finite storage and often must learn incrementally without offloading data (for privacy or connectivity reasons), (ii) *privacy*: prior task data may contain sensitive information that cannot be archived or sent to a server. These real-world constraints motivate our setting where the agent must learn sequentially **without full past data. On-device online MBRL** addresses scenarios where a pretrained universal simulator is unavailable or too large to deploy. DRAGO’s contribution is to some extent **complementary to foundation models**: for users who have access to a foundation model, our approach could be used to continually fine-tune that model on new tasks in a memory-efficient way. Conversely, in domains not covered by a foundation model (or where data cannot leave the device), DRAGO enables continual learning from scratch. We would also like to emphasize that even though large world model pretraining is very popular right now, it is not the only correct way and not sufficient to learn a perfect world model - we would definitely need more techniques (especially continual learning) in the future and we should not stop doing research on it.
>
> Q: Evaluation Procedure
>
> A: The “test tasks” in our evaluation are **new tasks presented to the agent after the sequence of training tasks**, meant to assess how well the learned world model can be reused. We apologize for the confusion – these test tasks do involve further RL training of the policy (i.e. the agent is still learning to maximize reward on the new task), and we only load the pretrained world model’s parameters during these tests. In other words, when the agent faces a test task, only the world model is loaded to **test if it is an ideal initialization **for the new task.
>
> Q: Integration with TD-MPC, Decoder, and Learner vs. Reviewer Agents
>
> A: *Decoder* usage: DRAGO is built on TD-MPC, but we introduce a **variational generative model** (encoder–decoder) for state-action pairs as part of our Synthetic Experience Rehearsal module (Section 3.1). TD-MPC by itself uses only an encoder and latent dynamics (planning entirely in latent space), so a decoder was not needed in the original TD-MPC. In our case, however, the **decoder is essential** – it allows us to **reconstruct synthetic state-action examples** from the latent generative model of past tasks. These reconstructed experiences are fed to the world model to rehearse past dynamics. In short, **without a decoder, the agent couldn’t simulate prior states/actions explicitly**, so we added one to enable **generative replay** of past experiences (we will clarify this design choice in the text).
>
> *Learner vs. Reviewer in planning:* In Section 3.2 and Algorithm 1, we introduce two parallel actor-critic pairs – a **“learner” agent** (the original policy optimizing extrinsic reward) and a **“reviewer” agent** (an auxiliary policy optimizing an intrinsic reward)​. Both **share the same world model** and run simultaneously during training, but they have separate policy networks and reward/value heads. The learner uses the environment’s reward $r^e$ to solve the task (just like standard TD-MPC), while the reviewer receives a designed intrinsic reward $r^i$ (Equation 7 in the paper) that encourages revisiting state transitions that the **previous task’s model** could predict confidently. In practice, the two agents alternate or parallelize interactions with the environment in each training iteration (we will clarify this scheduling in the revision). During planning for action selection, each agent uses Model Predictive Control (CEM in our case) with its own reward model: the learner plans to maximize extrinsic return, while the reviewer plans to maximize intrinsic return. They do not directly interfere with each other’s action selection; they simply contribute different trajectories for training.
>
> *Why not a single combined reward/value?* We chose to keep intrinsic and extrinsic rewards separate (two policies) after initial experiments indicated that combining them can be counterproductive. A single policy optimizing a sum of extrinsic+intrinsic rewards tended to trade off one against the other, sometimes neglecting the task objective in favor of curiosity (or vice versa). By using a dedicated reviewer agent for the intrinsic objective, we ensure the extrinsic task performance remains the learner’s sole focus, while the reviewer safely explores for retention. The world model benefits from both data sources without the learner’s policy being distracted.

---

> > ### Comment · Reviewer_5Rrm · 2025-04-02
> >
> > Thank you for your rebuttal. You have addressed many of my concerns.
> >
> > However, my concern about the choice of tasks remains unaddressed. I am not certain that these tasks have not been cherry-picked to demonstrate the success of this method. Can the authors comment on the choice of continual RL tasks and whether they have appeared in previous literature?
> >
> > My concerns here are slightly mitigated by the additional baselines from other methods that the authors are preparing for Reviewer XZ8k, so I am minded to increase my score if the authors can provide some more justification and / or results on a wider range of task orderings.

---

> > > ### Author Response · Authors · 2025-04-03
> > >
> > > Thank you for the follow-up comment. We appreciate your continued engagement with our work. We would like to emphasize that each individual tasks, i.e., MiniGrid goal reaching, Cheetah/Walker run, jump etc. , is one of the most common standard RL tasks as used in many well-known papers, like TDMPC, which is the MBRL algorithm that we built DRAGO on.
> > >
> > > Specifically, for MiniGrid: We chose distinct rooms in MiniGrid because each room highlights a different region of the state space, yet all rooms share underlying transition dynamics. Although the tasks are laid out in a way to ensure minimal overlap in reward-relevant regions, the “door-connecting” layout is standard in many MiniGrid experiments, and we believe it realistically captures cases where local behaviors (e.g., navigating a specific room) must be stitched together across tasks.
> > >
> > > DeepMind Control Suite (Walker, Cheetah): These tasks are standard continuous-control benchmarks. Our continual-learning versions simply vary reward functions (e.g., running vs. jumping vs. walking backward), which is a common way to induce different behavioral modes while preserving the same underlying physics. As we focus on knowledge retention in this paper, we design tasks like jump&run and jump2run, to make sure we can test in the same time whether the agent forgets previous knowledge and whether the agent is learning a more and more complete world model in the process of continual learning (so it can quickly solve the combination of previous tasks).
> > >
> > > Overall, we designed the tasks to (a) ensure that each new task reveals a different aspect of the dynamics (rather than reusing the same states or transitions repeatedly), and (b) require knowledge retention across tasks for better performance. Our main goal was not to artificially inflate our method’s advantages, but rather to demonstrate it on tasks that are not trivially overlapping.
> > >
> > >
> > > As the reviewer also mentioned, we have included two new baselines and have updated the results in the anonymous link: https://drive.google.com/file/d/18TdI9nPCT7MMQCQMmL1ZYxJkblhUI91i/view?usp=sharing. We compared to one replay-based MBRL baseline and one pseudo-rehearsal MBRL baseline, and DRAGO still clearly outperforms both, indicating that the combination of synthetic experience rehearsal and targeted intrinsic exploration is crucial for robust knowledge retention.

---

### Decision · Program_Chairs · 2025-05-01

**Decision:**

Accept (poster)

**Comment:**

This paper proposes an algorithm built on top of TDMCP which learns a generative model of prior environment experience and an exploration objective based on revisiting prior memories, for the ultimate objective on improving on catastrophic forgetting in continual learning settings. This is an important setting and the contribution here seems well-motivated and well-justified through experimentation: limited memory (i.e., as opposed to doing pure in-context learning) and privacy are important aspects to touch on, and the results are strong.

There were concerns on benchmarks, for example the environments being too limited or cherry picked and the baselines being too limited, but these seem to have been addressed in the rebuttal. There also seemed to be various concerns about clarity, but from reading the responses I believe these were sufficiently addressed, given that the final manuscript contains these clarifications.

I therefore recommend acceptance to the conference.